# Structure, function and assembly of soybean primary cell wall cellulose synthases

Ruoya Ho[1†], Pallinti Purushotham[1†‡], Louis FL Wilson[1,2], Yueping Wan[1], Jochen Zimmer[1,2]*

[1]Department of Molecular Physiology and Biological Physics, University of Virginia School of Medicine, Charlottesville, United States; [2]Howard Hughes Medical Institute, Chevy Chase, United States

## eLife Assessment

It is well established that cellulose synthesis in higher plants requires three different but related cellulose synthase (CESA) isoforms. Here the authors provide **convincing** biochemical and cryo electron microscopy structural information on the interactions within soybean primary cell wall CESA homotrimers. They present an **important** model in which multi-subunit cellulose synthase complexes are made of homotrimers of different CESA isoforms.

*For correspondence:
jz3x@virginia.edu

[†]These authors contributed equally to this work

Present address: [‡]Department of Life Sciences, GITAM University, Bengaluru, India

Competing interest: The authors declare that no competing interests exist.

**Abstract** Plant cell walls contain a meshwork of cellulose fibers embedded into a matrix of other carbohydrate and non-carbohydrate-based biopolymers. This composite material exhibits extraordinary properties, from stretchable and pliable cell boundaries to solid protective shells. Cellulose, a linear glucose polymer, is synthesized and secreted across the plasma membrane by cellulose synthase (CesA), of which plants express multiple isoforms. Different subsets of CesA isoforms are necessary for primary and secondary cell wall biogenesis. Here, we structurally and functionally characterize the *Glycine max* (soybean) primary cell wall CesAs CesA1, CesA3, and CesA6. The CesA isoforms exhibit robust in vitro catalytic activity. Cryo-electron microscopy analyses reveal their assembly into homotrimeric complexes in vitro in which each CesA protomer forms a cellulose-conducting transmembrane channel with a large lateral opening. Biochemical and co-purification analyses demonstrate that different CesA isoforms interact in vitro, leading to synergistic cellulose biosynthesis. Interactions between CesA trimers are only observed between different CesA isoforms and require the class-specific region (CSR). The CSR forms a hook-shaped extension of CesA's catalytic domain at the cytosolic water-lipid interface. Negative stain and cryo-electron microscopy analyses of mixtures of different CesA isoform trimers reveal their side-by-side arrangement into loose clusters. Our data suggest a model by which CesA homotrimers of different isoforms assemble into cellulose synthase complexes to synthesize and secrete multiple cellulose chains for microfibril formation. Inter-trimer interactions are mediated by fuzzy interactions between their CSR extensions.

## Introduction

Cellulose is a versatile biopolymer and a fundamental building block of plant cell walls. It is an amphipathic linear β–1,4 linked glucose polymer that can be assembled into fibrillar structures. As the load-bearing wall component of vascular plants, cellulose microfibrils are spun around the cell and integrated with a variety of other biopolymers (*Turner and Kumar, 2018*). Cellulose is synthesized by cellulose synthase (CesA), a membrane-integrated processive family-2 glycosyltransferase (GT; *Lombard et al.,*

*2014*). The enzyme synthesizes cellulose from UDP-activated glucose (UDP-Glc) and translocates the polymer across the plasma membrane through a channel formed by its own membrane-spanning segment (*McNamara et al., 2015*; *Morgan et al., 2013*). While bacterial CesAs primarily function as monomeric enzymes (*Abidi et al., 2021*; *Acheson et al., 2021*; *Du et al., 2016*; *Morgan et al., 2013*), structural analyses of plant CesAs revealed their assembly into triangular-shaped trimeric complexes of three catalytically active subunits (*Massenburg et al., 2024*; *Purushotham et al., 2020*; *Zhang et al., 2021a*).

The overall CesA architecture and the mechanism of cellulose biosynthesis is evolutionarily conserved from bacteria to land plants. However, plant CesAs contain specific domains absent in most bacterial homologs. These include an extended cytosolic N-terminus beginning with a RING-like region, a plant conserved region (PCR), as well as a class-specific region (CSR; *Purushotham et al., 2020*). The CSR and PCR are inserted into the cytosolic catalytic domain. While the PCR is a trimerization domain that connects three CesA promoters in a trimeric complex, the function and structure of the CSR remains unknown. Further, plants express different CesA isoforms at different developmental stages, of which certain subsets are necessary for primary and secondary cell wall formation (*Persson et al., 2007*; *Taylor et al., 2003*; *Turner and Somerville, 1997*). Based on the *Arabidopsis* nomenclature, isoforms associated with primary cell wall formation include CesA1, CesA3, and CesA6, whereas secondary cell wall CesAs are CesA4, CesA7, and CesA8. The isoenzymes vary the most within the CSR and the N-terminal domain.

Genetic analyses demonstrated the importance of different CesA isoforms for primary and secondary cell wall formation (*Fagard et al., 2000*; *Turner and Somerville, 1997*; *Persson et al., 2007*; *Sampathkumar et al., 2019*). Further, co-immunoprecipitation analyses indicated direct interactions between primary or secondary cell wall CesA isoforms (*Gonneau et al., 2014*; *Taylor et al., 2003*; *Timmers et al., 2009*). However, the specific functions of the different CesA isoforms during in vivo cellulose biosynthesis remain unknown.

Plants organize cellulose into micro and macro-fibrils (*Kubicki et al., 2018*; *Turner and Kumar, 2018*). Cellulose microfibrils likely originate from supramolecular CesA complexes (CSCs) observed in various species. In land plants, CSCs appear primarily as pseudo sixfold symmetric membrane-integrated clusters by freeze fracture electron microscopy analyses (*Herth and Weber, 1984*; *Kimura et al., 1999*; *Nixon et al., 2016*). The CesA trimer likely represents the CSC repeat unit, thereby accounting for 18 CesAs per CSC and, accordingly, 18 cellulose polymers in a CSC-synthesized microfibril (*Cosgrove et al., 2024*; *Nixon et al., 2016*; *Turner and Kumar, 2018*).

To analyze the oligomerization and function of primary cell wall CesAs, we recombinantly expressed and purified *Glycine max* (soybean, Gm) CesA1, CesA3, and CesA6. Cryo-EM analyses of all three CesA isoforms reveal the formation of homotrimeric complexes, similar to the secondary cell wall CesAs. The CSR is resolved at the corners of the CesA trimer as a disordered but hook-shaped domain that runs at the cytosolic water-lipid interface. In vitro co-purification and electron microscopy studies demonstrate that homotrimers of different CesA isoforms interact. This interaction requires the CSR and leads to synergistic cellulose biosynthesis. Our results support a model by which CSCs are formed from homotrimers of different CesA isoforms.

## Results
### Primary cell wall CesAs purify as high and low molecular weight species

We selected a set of *G. max* (*Gm*) CesA isoforms that phylogenetically cluster with *Arabidopsis thaliana* CesA1, CesA3 and CesA6, respectively, and that are widely and strongly expressed in unlignified soybean tissues (*Figure 1*, *Figure 1—figure supplement 1A and B*). Existing co-expression data from ATTED-II https://atted.jp/ (*Obayashi et al., 2022*) indicate that the enzymes are co-expressed with other primary cell wall genes implicated in pectin, arabinogalactan, and galactoglucomannan biosynthesis (*Figure 1*). We therefore conclude that the selected CesAs indeed represent primary cell wall CesAs.

To biochemically and structurally characterize the *Gm*CesAs, we followed a similar heterologous expression protocol as established previously for hybrid aspen CesA8 (*Ptt*CesA8) (*Purushotham et al., 2020*). In short, the *Gm*CesAs were expressed with N-terminal poly-histidine tags in Sf9 insect cells and purified by metal affinity and size exclusion chromatography in the detergent glyco-diosgenin (GDN;

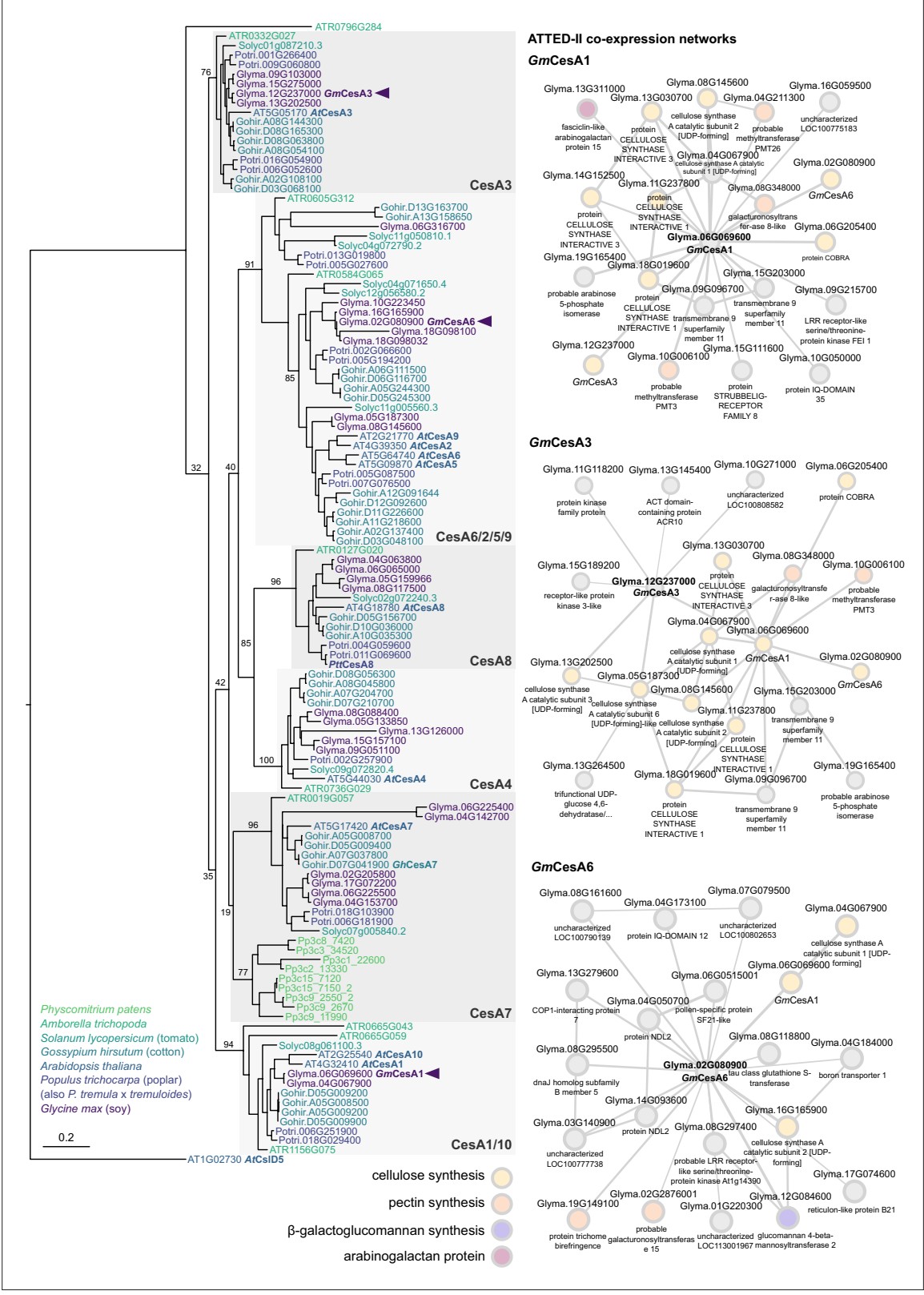

**Figure 1.** *Gm*CesA phylogeny and co-expression analysis. Left: Maximum-likelihood phylogeny of CesA protein sequences from soy, *Arabidopsis*, cotton, poplar, tomato, *Physcomitrium* and *Amborella*. For alignment, the Pfam-defined 'Cellulose synthase' domain was extracted from each sequence using HMMER. The final phylogeny was calculated using RAxML with 100 rapid bootstrap pseudo-replicates. Arrowheads mark the positions of *Gm*CesA1, *GmCesA3*, and *Gm*CesA6 within the tree; structurally characterized proteins (*Ptt*CesA8 and *Gh*CesA7) and *Arabidopsis* sequences are also

*Figure 1 continued on next page*

*Figure 1 continued*

labelled. Branch lengths correspond to average number of substitutions per site (relative to scale bar); branch labels report bootstrap successes for each split. Right: Co-expressed gene networks for *Gm*CesA1, *Gm*CesA3, and *Gm*CesA6 from ATTED-II v11 (https://atted.jp). Relevant functional annotations for co-expressed cell wall genes are labelled by color (cellulose synthesis: yellow; pectin synthesis: orange; β-galactoglucomannan synthesis: violet; arabinogalactan proteins: purple).

The online version of this article includes the following figure supplement(s) for figure 1:

**Figure supplement 1.** Sequence alignment of soybean CesA1, CesA3, and CesA6 and substrate turnover kinetics.

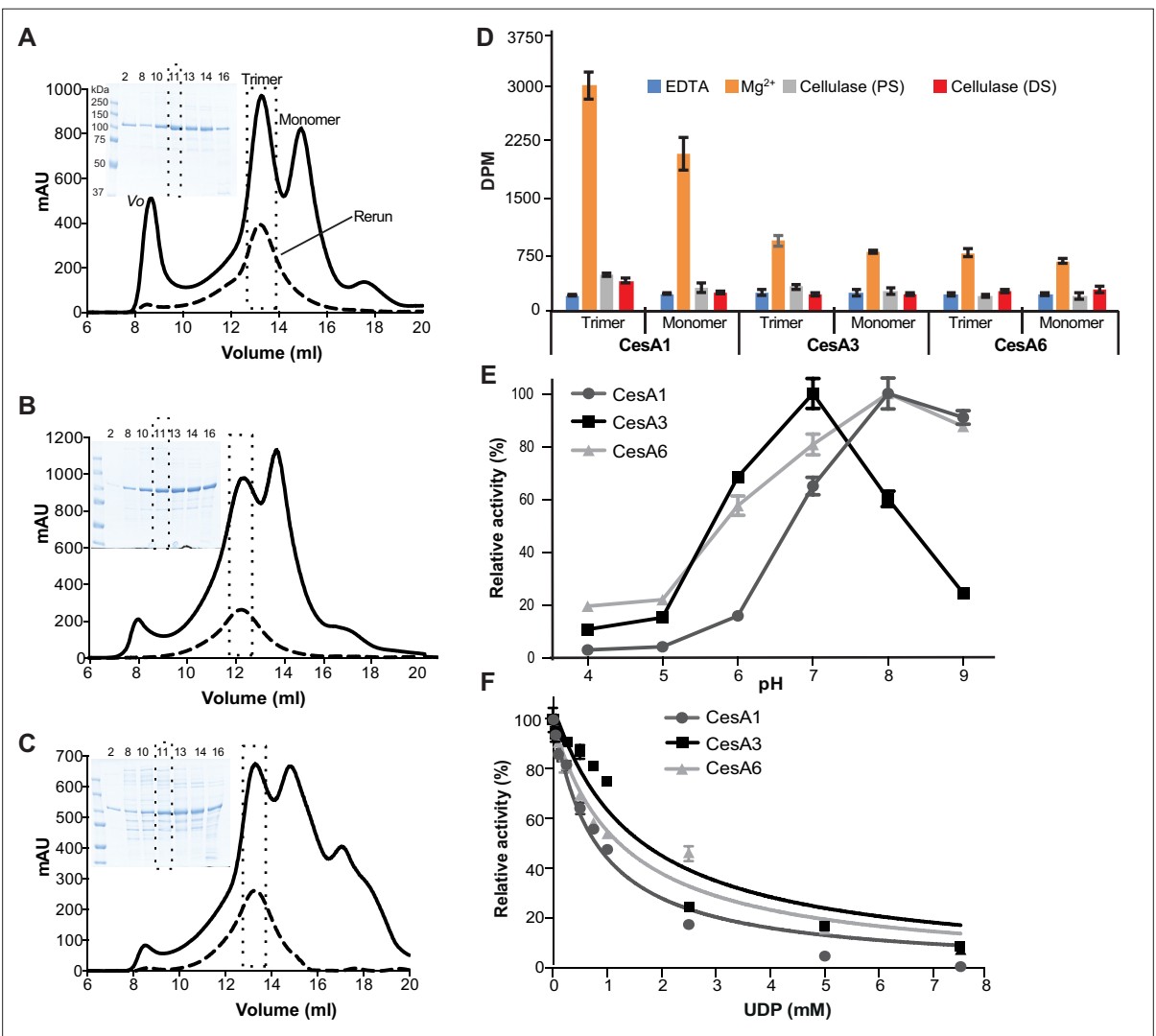

**Figure 2.** Functional characterization of *Glycine max* primary cell wall CesAs. From (**A–C**) analytical size exclusion chromatography (Superose 6 Increase) of *Gm*CesA1 (**A**), *Gm*CesA3 (**B**), and *Gm*CesA6 (**C**). Void volume (*Vo*) and trimer and monomer peaks are marked. A rerun of the trimer fraction for each species is shown as a dashed profile. Inset: Coomassie-stained SDS-polyacrylamide gel electrophoresis of the indicated elution volumes. The molecular weights of the protein marker bands are indicated in panel A and apply to all panels. (**D**) Catalytic activity of the purified *Gm*CesAs. $^3$H-labeled cellulose synthesized by trimeric and monomeric species was degraded with cellulase, followed by quantification by scintillation counting. (DS) and (PS) indicate cellulase treatments during and after the synthesis reaction, respectively. DPM: disintegrations per minute. (**E**) pH optima for catalytic activity of *Gm*CesA1, *Gm*CesA3, and *Gm*CesA6. Activities are normalized to the highest activity for each isoform. (**F**) UDP inhibits CesAs. Cellulose biosynthesis was performed in the presence of 1.4, 0.5, and 2.3 mM UDP-Glc for *Gm*CesA1, *Gm*CesA3, and *Gm*CesA6, respectively, as well as the indicated increasing concentrations of UDP. Product yields in the absence of UDP were set as 100%. Error bars in panels D–F represent deviations from the means of at least three replicas.

Materials and methods). Size exclusion chromatography separated all *Gm*CesA isoforms into high and low molecular weight species (*Figure 2A–C*). Cryogenic and negative stain EM analyses identified these species as *Gm*CesA trimers and monomers, respectively (see below). Of note, compared to *Gm*CesA1 and *Gm*CesA3, the yield of trimeric *Gm*CesA6 was more variable, with some preparations producing primarily monomeric species. This suggests that *Gm*CesA6 is less stable in a detergent-solubilized state compared with the other isoforms.

To test whether the detergent-solubilized *Gm*CesA trimers dissociate into monomers over time, the purified *Gm*CesA trimers were reinjected onto the size exclusion chromatography column after an overnight incubation on ice. For all species, the reinjected material eluted as a trimeric complex, indicating that assembled trimers are stable and do not interconvert with monomers within this time-frame (*Figure 2A–C*). The co-purifying monomers likely arise from incompletely assembled trimers or oligomer dissociation during purification.

## In vitro cellulose biosynthesis

Cellulose biosynthetic activity of the purified CesAs was quantified by measuring the incorporation of $^3$H-labeled glucose into insoluble cellulose, followed by scintillation counting, as previously described (*Purushotham et al., 2016*). As shown in *Figure 2D*, the relative activities of the monomeric and trimeric *Gm*CesA fractions are comparable for each isoform, demonstrating that both species are catalytically active in vitro. Between the different isoforms, *Gm*CesA1 exhibits greatest product accumulation (*Figure 2D*). In all cases, the in vitro synthesized polymer is readily degraded by a cellulase, indicating the formation of authentic cellulose. No product was obtained in the presence of EDTA, in agreement with previous observations (*Purushotham et al., 2016*).

To further assess catalytic differences between the *Gm*CesA isoforms, we determined their pH optima for catalytic activity (*Figure 2E*). All *Gm*CesA isoforms show greatest catalytic activity at neutral to mild-alkaline pH. *Gm*CesA3 exhibits an activity optimum at pH 7 with a sharp decline at pH 8 and 9. In contrast, the activities of *Gm*CesA1 and *Gm*CesA6 peak at pH 8, with a slight decline at pH 9 (*Figure 2E*). Quantifying the release of UDP during biosynthesis reactions using a 'UDP-Glo' glycosyltransferase assay (*Das et al., 2016*) reveals Michaelis–Menten constants for *Gm*CesA1 and *Gm*CesA3 with respect to UDP-Glc of 0.44 mM and 0.18 mM, respectively. The apparent $V_{max}$ value is ~eightfold higher for *Gm*CesA1 compared to *Gm*CesA3 (*Figure 1—figure supplement 1C*). The activity of *Gm*CesA6 was too weak to be analyzed by this method. Further, we analyzed inhibition of the isoforms by UDP, which competitively inhibits *Ptt*CesA8 and related GT-2 enzymes (*Gow and Selitrennikoff, 1984*; *Omadjela et al., 2013*; *Purushotham et al., 2016*; *Tlapak-Simmons et al., 2004*). *Gm*CesA1's apparent $IC_{50}$ for UDP is about 0.8 mM, whereas this concentration is increased to about 1.3–1.5 mM for *Gm*CesA6 and *Gm*CesA3, respectively (*Figure 2F*).

## Homotrimer assembly

Cryo-EM analyses of the high molecular weight *Gm*CesA fractions (*Figure 2A–C*) revealed their organization into homotrimeric complexes (*Figure 3A*, *Figure 3—figure supplements 1 and 2*, and *Supplementary file 1*). For all isoforms, two-dimensional classification identified trimeric particles similar to *Ptt*CesA8 and cotton CesA7 (*Gh*CesA7) (*Purushotham et al., 2020*; *Zhang et al., 2021a*). Particle classification in three dimensions followed by non-uniform refinement with applied C3 symmetry and local refinement generated cryo-EM maps ranging in resolution from about 3.0–3.3 Å.

Overall, the CesAs contain a cytosolic catalytic domain that interacts with the channel-forming TM region via three amphipathic IF helices (*Figure 3B*). The helices surround the entrance to the TM channel with the Trp residue of the QxxRW motif at its portal. As previously described for hybrid aspen *Ptt*CesA8 and *Gh*CesA7, the CesA trimers are stabilized by the PCR domain that is inserted into CesA's catalytic domain. (*Purushotham et al., 2020*; *Zhang et al., 2021a*).

The triangular PCR arrangement in a CesA complex positions the side chains of conserved Lys and Arg residues towards the threefold symmetry axis. These residues include Arg449, Lys452, and Arg453 in *Gm*CesA6 and coordinate unidentified ligand(s) on the membrane distal and proximal side of the PCR triangle. For all three CesA isoforms, the ligands' shapes are similar, suggesting that they represent the same small molecule (*Figure 3—figure supplement 2B*). On the membrane proximal side, the density extends by about 9 Å towards the membrane, perhaps representing a nucleotide bound in different poses, as previously suggested for *Ptt*CesA8 (*Purushotham et al., 2020*).

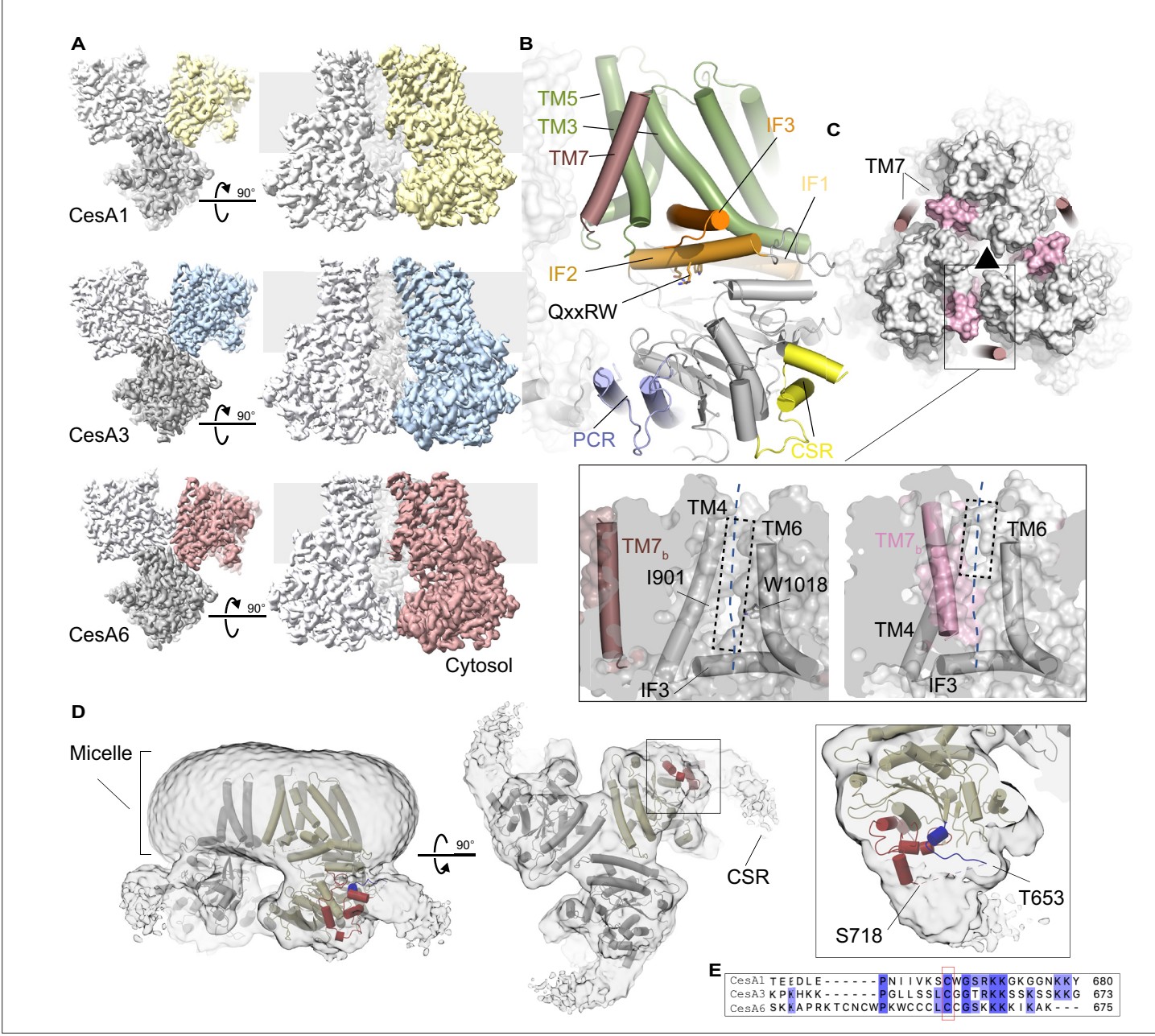

**Figure 3.** Soybean primary cell wall CesAs assemble into homotrimers. (**A**) CryoEM maps of the *Gm*CesA homotrimers contoured at 4.5–5.6 σ. One subunit is shown in color, the others are shown in light and dark gray. The gray background indicates the estimated membrane boundaries. (**B**) Cartoon representation of a *Gm*CesA6 protomer. The transmembrane region is shown in green and dark pink, interface helices (IF) are shown in orange, and the catalytic domain is colored gray. The PCR and CSR regions are shown in blue and yellow, respectively. (**C**) Comparison of *Gm*CesA6 and *Ptt*CesA8. *Gm*CesA6 is shown as a cartoon that is overlaid with a semitransparent surface of *Ptt*CesA8 (surface, PDB: 6WLB). Transmembrane helix 7 is colored light and dark pink for *Ptt*CesA8 and *Gm*CesA6, respectively. The black triangle indicates the threefold symmetry axis of the homotrimer. Zoom views: Surface representations of *Gm*CesA6 (left) and *Ptt*CesA8 (right) highlighting the lateral window. TM7_b refers to TM helix 7 of another protomer. The view is from the threefold symmetry axis towards a CesA protomer. The dashed blue line indicates the cellulose secretion channel. (**D**) CryoEM map of the *Gm*CesA1 trimer shown at a low contour level (1.4 σ). The *Gm*CesA1 structure is shown as a cartoon with one protomer colored yellow. The resolved CSR N- and C-terminal helical regions are colored blue and red, respectively. (**E**) Sequence alignment of the CSR regions of *Gm*CesA1, *Gm*CesA3, and *Gm*CesA6 generated in Clustal Omega (*Larkin et al., 2007*).

The online version of this article includes the following figure supplement(s) for figure 3:

**Figure supplement 1.** Cryo-EM data processing workflows.

**Figure supplement 2.** Cryo-EM map quality examples.

## A transmembrane channel with a large lateral opening

CesAs contain seven TM helices of which helices 1–6 create a cellulose conducting channel. In the previously described *Ptt*CesA8 and *Gh*CesA7 complexes, TM helix 7 of one protomer packs against TM helices 5 and 6 of a neighboring CesA subunit (*Figure 3C*). In new soybean CesA structures, however, this helix is more flexible, as evidenced by weaker map quality (*Figure 3—figure supplement 2A*), and shifted to the periphery of the trimer. In the new position, the helix primarily mediates contacts with TM helix 5 and the C-terminal segment of TM helix 3 of the same CesA protomer (*Figure 3B and C*). The helix has been modeled for *Gm*CesA1 and *Gm*CesA6, while its density is detectable at a similar position but too discontinuous for modeling in the *Gm*CesA3 map.

The displacement of TM helix 7 away from the TM channel of the neighboring subunit opens a lateral lipid-exposed window in the neighboring subunit's channel architecture (*Figure 3C*). The window is formed by the N-terminal region of IF helix 3 and TM helices 4 and 6. About midway across the membrane, the opening is roughly 6 Å wide, for example between the side chains of Ile901 in TM helix 4 and Trp1018 in TM helix 6 (*Figure 3C*). The lateral window likely exposes the translocating nascent cellulose polymer to the hydrophobicity of the lipid bilayer. A similar lipid exposed polysaccharide translocation pathway has recently been described for hyaluronan synthase (*Maloney et al., 2022*).

## The CSR forms a hook-shaped extension of the catalytic domain

Plant CesAs contain two structurally and functionally unresolved domains, which are the N-terminal domain (NTD) and the CSR. The NTD has been resolved at lower resolution for *Ptt*CesA8, where it forms a helical stalk extending from the catalytic domains into the cytosol (*Purushotham et al., 2020*). We observe a similar stalk-like extension in some of the trimeric *Gm*CesA3 particles (*Figure 3—figure supplement 1*). However, the NTD is only resolved for a small subset of *Gm*CesA3 particles and not resolved at all for the other *Gm*CesAs. This suggests that it can adopt multiple conformations, with the stalk being one of them.

The CSR has been proposed to be intrinsically disordered (*Scavuzzo-Duggan et al., 2018*) and only its short N- and C-terminal helical segments are visible in the cryo-EM maps. For *Gm*CesA1, however, at lower contour levels, additional CSR density is evident, extending from corners of the catalytic domains of the *Gm*CesA1 trimer (*Figure 3D*). Viewed from the cytosol, the extra density resembles a hook extending by about 20 Å clockwise and tangentially along the trimer's corners at the water-lipid interface. At this position, the CSR's N-terminal conserved cysteine residue(s) postulated to be acylated (*Kumar et al., 2016*) reside near the membrane interface (*Figure 3D and E*).

## Homotrimers of different *Gm*CesA isoforms interact

We next tested whether *Gm*CesA homotrimers of different isoforms would interact in vitro. To this end, we individually expressed and purified trimers of poly-His tagged *Gm*CesA1 and *Gm*CesA6 (His-CesA1 and His-CesA6) and TwinStrep-tagged *Gm*CesA1 and *Gm*CesA3 (Strep-CesA1 and Strep-CesA3) (*Figure 4*). Cross-isoform interactions were tested by tandem purifications over Ni-NTA and Strep-Tactin affinity matrices. His-CesA1 can be distinguished from Strep-*Gm*CesA3 by Coomassie stained SDS-PAGE due to size differences, whereas all other species comigrate. Therefore, western blotting together with Coomassiestained SDS-PAGE was performed to evaluate the co-purification results.

An equimolar mixture (based on UV absorbance) of His-CesA1 and Strep-CesA3 was incubated for 180 min on ice and sequentially purified using (1) Ni-NTA resin and (2) Strep-Tactin beads (*Figure 4A*). The Coomassie stained SDS-PAGE resolved both *Gm*CesA species in the initial mixture, after elution from the Ni-NTA resin, as well as upon elution from the Strep-Tactin beads. The identity of the bands as His-CesA1 and Strep-CesA3 was confirmed by western blotting. We observed no non-specific binding of His-CesA1 to Strep-Tactin beads or Strep-CesA3 to Ni-NTA resin (*Figure 4E*).

Similar experiments with combinations of Strep-CesA3 and His-CesA6 (*Figure 4B*) and Strep-CesA1 and His-CesA6 (*Figure 4C*) yielded comparable results, although the co-eluting species cannot be distinguished by Coomassie staining alone, due to comigration. None of the species showed detectable non-specific binding to the affinity resins in the absence of the corresponding tags (*Figure 4E–G*).

As an additional control, we analyzed whether differently tagged homotrimers of the same isoform also interact with each other. To this end, a mixture of His-CesA1 and Strep-CesA1 was subjected to

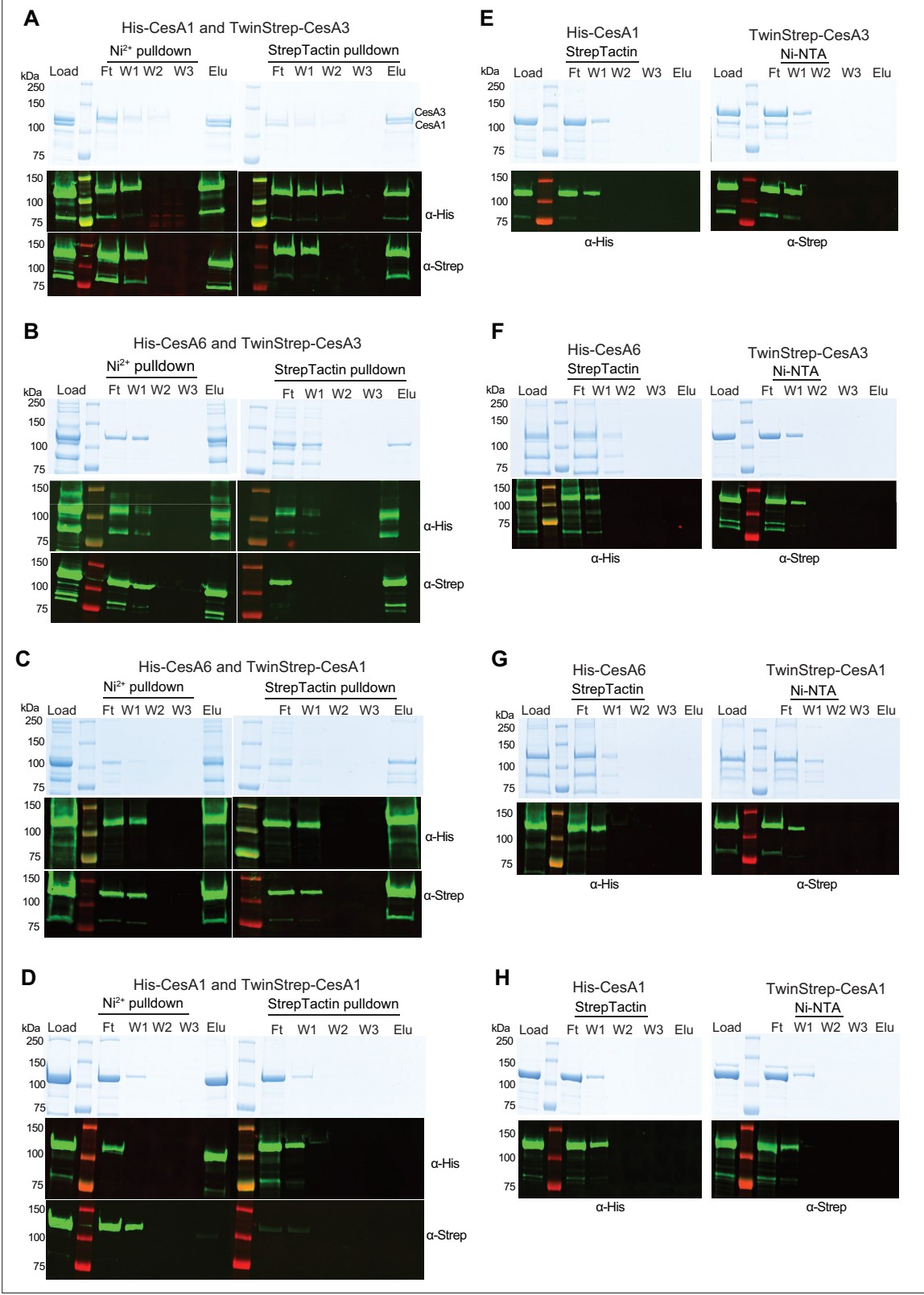

**Figure 4.** In vitro interactions between different CesA isoforms. Tandem pull-down experiments using Ni-NTA and Strep-Tactin resin. Experiments were performed with homotrimers of the indicated *Gm*CesA isoforms tagged N-terminally either with His- or TwinStrep-tags. Material eluted from the Ni-NTA resin was loaded onto the Strep-Tactin beads. Top panels: Coomassie stained SDS-PAGE, bottom panels: Western blots using anti penta-His or anti-Strep primary antibodies. (**A–C**) Trimer-trimer interaction between *Gm*CesA1 and *Gm*CesA3, *Gm*CesA6 and *Gm*CesA3, and *Gm*CesA6 and *Gm*CesA1,

*Figure 4 continued on next page*

*Figure 4 continued*

respectively. (**D**) Differently tagged homotrimers of the same isoform do not interact. Tandem purification of a mixture of His- and TwinStrep-tagged *Gm*CesA1. (**E–H**) Control binding of His-tagged CesAs to StrepTactin beads and Strep-tagged *Gm*CesAs to Ni-NTA resin. F, W, E: Flow through, wash, and eluted fractions.

The online version of this article includes the following source data and figure supplement(s) for figure 4:

**Source data 1.** Raw uncropped data of western blots and Coomassie-stained PAGE gels shown in *Figure 4A-H*.

**Source data 2.** Boxed source data of western blots and Coomassie-stained PAGE gels shown in *Figure 4A-H*.

**Figure supplement 1.** Interactions of CesA homotrimers of the same isoforms.

**Figure supplement 1—source data 1.** Raw uncropped data of western blots and Coomassie-stained PAGE gels shown in *Figure 4—figure supplement 1A-F*.

**Figure supplement 1—source data 2.** Boxed source data of western blots and Coomassie-stained PAGE gels shown in *Figure 4—figure supplement 1A-F*.

**Figure supplement 2.** Attempt to purify hetero-oligomeric CesA oligomers.

**Figure supplement 2—source data 1.** Raw uncropped data of western blots and Coomassie-stained PAGE gels shown in *Figure 4—figure supplement 2D*.

**Figure supplement 2—source data 2.** Boxed source data of western blots and Coomassie-stained PAGE gels shown in *Figure 4—figure supplement 2D*.

tandem affinity purification as described above (*Figure 4D*). We failed to detect any co-purification of Strep-CesA1 when applying Ni-NTA as the first affinity chromatography step. Similar results were obtained for Strep- and His-tagged combinations of homotrimers of CesA3 or CesA6 (*Figure 4—figure supplement 1A–D*). Qualitatively, our interaction data are consistent with previously published co-immunoprecipitations of primary and secondary CesA isoforms (*Gonneau et al., 2014*; *Timmers et al., 2009*).

Similar tandem purification experiments were also performed with the monomeric CesA fractions obtained from size exclusion chromatography (*Figure 2A–C*). As observed for the homotrimeric complexes, monomeric His-CesA1 co-purifies with monomeric Strep-CesA3, demonstrating that trimeric assemblies are not necessary for the observed interactions (*Figure 4—figure supplement 1E and F*).

## Only the CSR is required for isoform interaction

The biological functions of CesA's NTD and CSR are currently unknown. The NTD's RING-like domain has been shown to form dimers and trimers in vitro (*Kurek et al., 2002*; *Purushotham et al., 2020*), raising the possibility that it could form inter-trimer complexes accounting for the observed isoform interactions.

To test this hypothesis, N-terminally truncated constructs of *Gm*CesA1 and *Gm*CesA3 were expressed and purified as described for the full-length variants. The constructs lack the first 259 (*Gm*CesA1) and 242 (*Gm*CesA3) residues yet purify as trimers (besides monomers) and exhibit in vitro catalytic activity similar to the full-length constructs (*Figure 5A*). A tandem affinity purification of a mixture of the truncated His-CesA1 and Strep-CesA3 isoforms demonstrates their interaction in the absence of the NTD, similar to the full-length enzymes (*Figure 5B and C* and *Figure 4A*).

To test whether the CSR is involved in isoform interaction, the domain was replaced in the N-terminally truncated *Gm*CesA1 and *Gm*CesA3 constructs with a flexible loop of 20 residues (see Materials and methods), thereby generating *Gm*CesA constructs devoid of the NTD as well as the CSR (*Figure 5D*). Negative stain EM of the purified truncated *Gm*CesA1 particles demonstrates their trimeric assembly, as observed for the full-length and NTD truncated versions (*Figure 5D*). Upon deletion of the CSR, the *Gm*CesA1 and *Gm*CesA3 isoforms no longer co-purify in vitro, suggesting that the region is indeed required for inter-isoform interaction (*Figure 5E*). Further, an N-terminally and CSR truncated *Gm*CesA3 construct shows only minor, most likely non-specific, interaction with full-length *Gm*CesA1, suggesting that *Gm*CesA isoforms interact primarily via their CSR domains (*Figure 5F*).

## Clustering of homotrimers of different *Gm*CesA isoforms

Negative stain electron microscopy analyses of the individual *Gm*CesA homotrimers revealed monodisperse particle distributions (*Figure 6A–C*). The size and shape of the particles is consistent with the

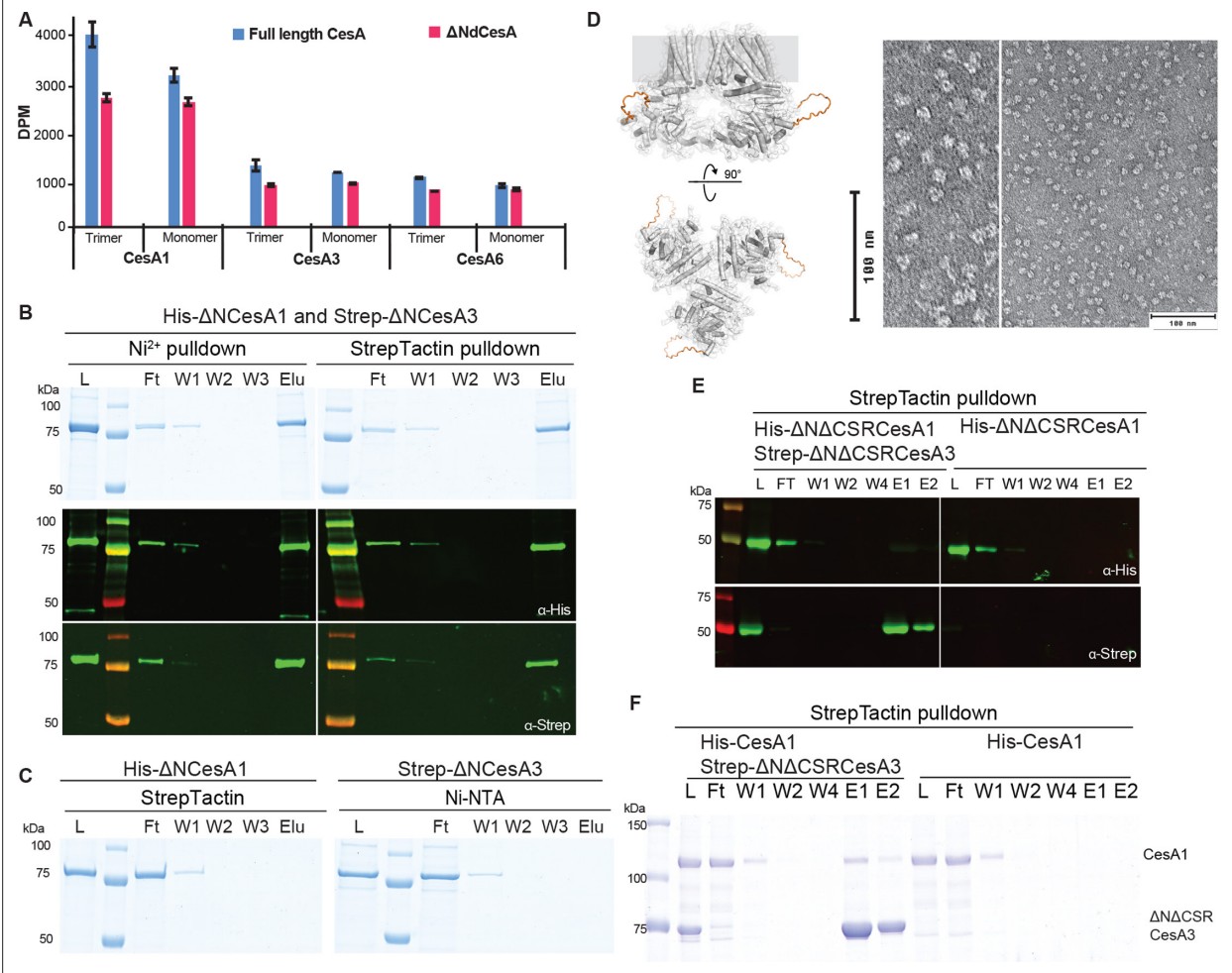

**Figure 5.** The CSR mediates trimer-trimer interactions. Isoform interactions are independent of the NTD. (**A**) Activity comparison of full-length and N-terminally truncated *Gm*CesA isoforms. DPM: disintegrations per minute. Error bars represent deviations from the means of at least three replicates. (**B**) Tandem pull-down experiments as in *Figure 4* but with N-terminally truncated homotrimers of *Gm*CesA1 and *Gm*CesA3. Top panel: SDS-PAGE, bottom panel: Western blots using anti-His and anti-Strep primary antibodies. (**C**) Control binding of His-ΔNCesA1 to Strep-Tactin beads and Strep-ΔNCesA3 to Ni-NTA resin. L, Ft, W, E: Load, Flow through, Wash, and Eluted fractions. (**D**) AlphaFold predicted model of N-terminally truncated *Gm*CesA1 with the CSR replaced by a loop shown as an orange backbone (left), and negative stain images of the NTD- and CSR-truncated *Gm*CesA1 trimer. (**E**) Tandem purification of NTD and CSR truncated *Gm*CesA1 and *Gm*CesA3. (**F**) Affinity purification of His-tagged full-length *Gm*CesA1 and Strep-tagged NTD- and CSR-truncated *Gm*CesA3. Shown is an SDS-PAGE after Coomassie staining.

The online version of this article includes the following source data for figure 5:

**Source data 1.** Raw uncropped data of western blots and Coomassie-stained PAGE gels shown in *Figure 5B, C, E and F*.

**Source data 2.** Boxed source data of western blots and Coomassie-stained PAGE gels shown in *Figure 5B, C, E and F*.

*Gm*CesA homotrimers observed by cryo-EM and remains unchanged over a course of a week when incubated on ice.

To visualize the interaction of the isoforms, all three homotrimeric isoforms were combined at equal molar ratio and subjected to size exclusion chromatography after incubation overnight on ice. Negative stain EM analysis of high molecular weight fractions eluting after the void volume revealed *Gm*CesA clusters of varying stoichiometries, ranging from 2 to >10 particles (*Figure 6D*). The clusters likely arise from trimer-trimer interactions in different orientations and vary in diameter from about 50–100 nm. No clustering was observed for any of the individual *Gm*CesA isoforms alone, even after prolonged incubations on ice.

Further, cryo-EM was used to analyze a mixture of individually purified *Gm*CesA1 and *Gm*CesA3 trimers. The trimers were combined and incubated on ice for 60 min prior to cryo grid preparation.

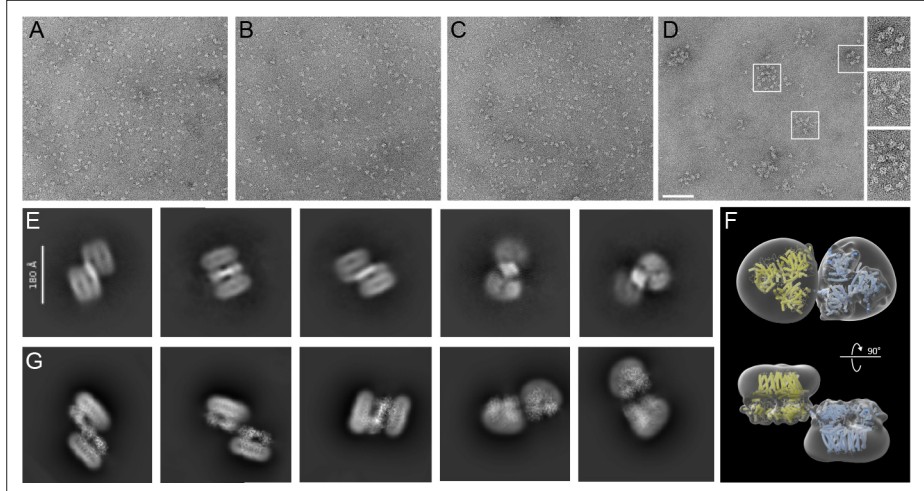

**Figure 6.** Clustering of *Gm*CesA homotrimers and a dimer of trimer. (**A–C**) Uranyl formate-stained EM images of homotrimers of purified *Gm*CesA1 (**A**), *Gm*CesA3 (**B**) and *Gm*CesA6 (**C**). The proteins were incubated overnight on ice prior to grid preparation. (**D**) The same for an equimolar mixture of all three *Gm*CesA isoforms, incubated overnight, separated from individual trimers by size exclusion chromatography, and imaged by negative stain EM. Selected clusters are encircled. Scale bar: 100 nm. (**E**) Purified trimers of *Gm*CesA1 and *Gm*CesA3 were combined and used for cryo-EM analysis. Shown are 2D class averages of dimers of trimers. (**F and G**) Manually assembled *Gm*CesA1 trimer volumes (**F**) were used to calculate 2D class average templates (**G**) for comparison with the experimentally obtained dimers of trimers shown in panel (**E**).

Focusing on particles larger than a CesA trimer, two-dimensional classification of the obtained particles revealed shapes resembling arrangements of two *Gm*CesA trimers. The identified particles are consistent with side-by-side arrangements of two *Gm*CesA trimers in opposite orientations. By this organization, the catalytic domains of the trimers are co-planar and the micelle-embedded TM segments are above and below the plane (***Figure 6E***).

To support this interpretation, two cryo-EM volumes of a *Gm*CesA1 trimer were manually arranged side-by-side but in inverted orientations. The individual volumes were placed such that the CSR densities of two CesA subunits of each trimer would contact each other. The obtained dimer-of-trimer volume was then used to calculate two-dimensional projections for comparison with the experimentally obtained 2D class averages. Indeed, the generated 'upside down and side-by-side' arrangement of *Gm*CesA trimer volumes resembles the experimental class averages (***Figure 6F and G***). Steric interferences of the micelle-embedded TM regions likely prevent the physiological parallel arrangement of two *Gm*CesA trimers under the experimental conditions.

## Synergistic cellulose biosynthesis

We investigated whether the cross-isoform interaction of *Gm*CesA trimers affects their in vitro catalytic activities. To this end, in vitro cellulose biosynthesis was quantified radiometrically from reactions containing one *Gm*CesA isoform at a constant concentration (20 µM) and increasing concentrations of a different *Gm*different CesA (1–20 µM). As a reference, cellulose biosynthetic activities were also determined for each isoform alone at the concentrations used in the combined assays. As shown in ***Figure 7A–F***, for all isoform combinations, the measured activities exceed the theoretical activities (calculated by adding the individually measured activities) at least one to two-fold, depending on the isoform combination. This suggests synergistic cellulose biosynthesis in the presence of two CesA isoforms. Performing the titration experiment with samples of the same isoform does not reveal any synergy, consistent with the lack of interaction between trimers of the same CesA isoforms (***Figures 7G and 4D*** and ***Figure 7—figure supplement 1***).

Additionally, comparing the measured and additive activities obtained after combining all three *Gm*CesA isoforms (at 6.6 µM each) reveals an experimental activity about threefold above the additive value. This activity level may arise from different dimeric arrangements of *Gm*CesA trimers (1+3, 1+6, and 3+6) and/or the formation of larger complexes of different isoform trimers (1+3 + 6; ***Figure 7H***).

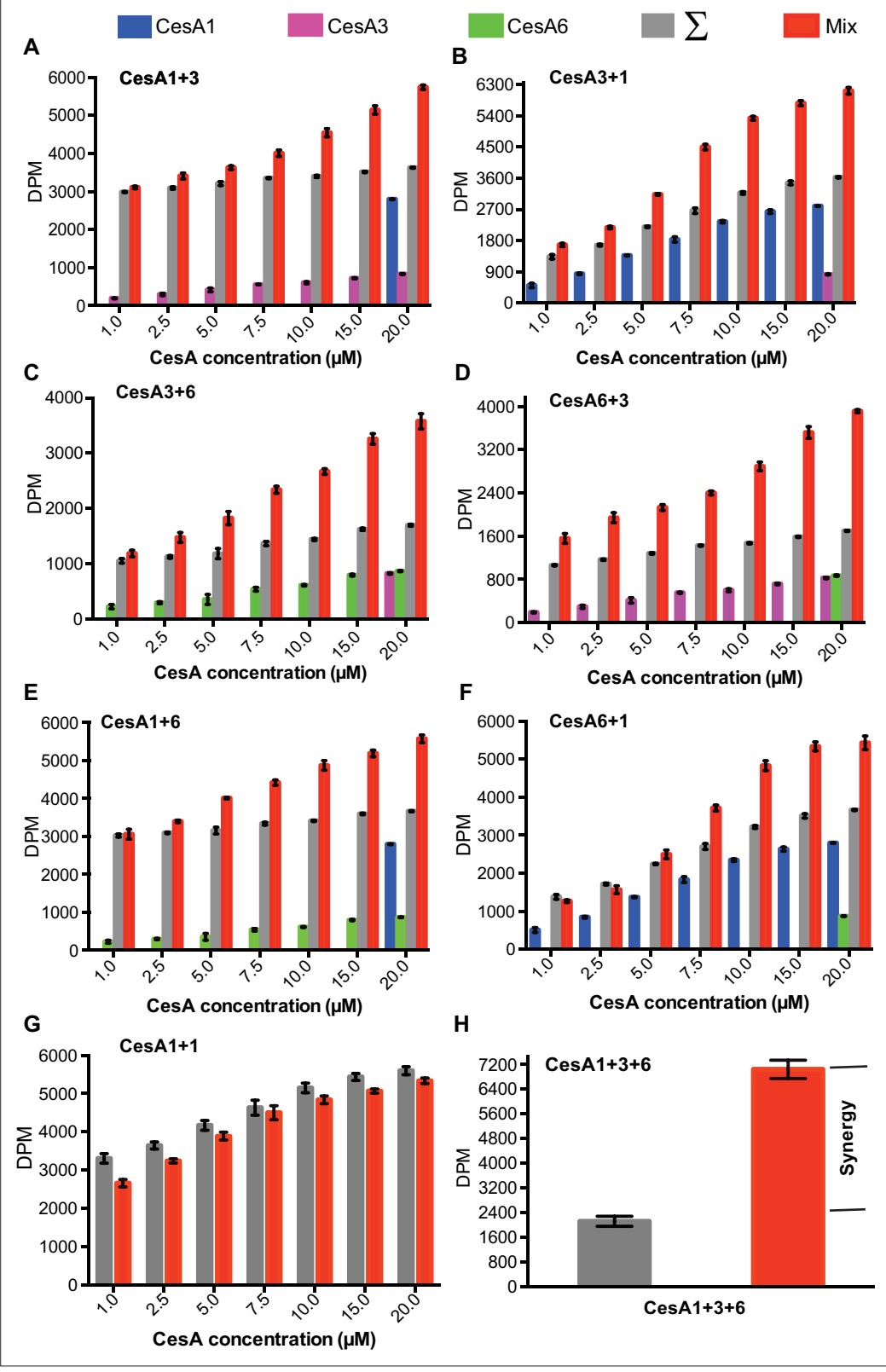

**Figure 7.** Synergistic catalytic activity. Cellulose biosynthesis from mixtures of *Gm*CesA isoform homotrimers. The formation of ³H-labeled cellulose was quantified by scintillation counting for reaction mixtures containing one *Gm*CesA isoform at 20 µM concentration and another isoform at the indicated increasing concentrations. Blue, magenta and green columns represent activities measured for the individual single isoforms alone. Gray columns

*Figure 7 continued on next page*

*Figure 7 continued*

represent the calculated theoretical activities for the isoform mixtures by adding the individually determined activities. Red columns represent the experimentally determined activities for the isoform mixtures. (**A and B**) CesA1$_{20\,\mu M}$ + CesA3$_{1-20\,\mu M}$ and CesA3$_{20\,\mu M}$ + CesA1$_{1-20\,\mu M}$; (**C and D**) CesA3$_{20\,\mu M}$ + CesA6$_{1-20\,\mu M}$ and CesA6$_{20\,\mu M}$ + CesA3$_{1-20\,\mu M}$; and (**E and F**) CesA1$_{20\,\mu M}$ + CesA6$_{1-20\,\mu M}$ and CesA6$_{20\,\mu M}$ + CesA1$_{1-20\,\mu M}$, respectively. (**G**) The same as for panel (**A**) but titrating the same *Gm*CesA isoform (CesA1$_{20\,\mu M}$ + CesA1$_{1-20\,\mu M}$). (**H**) The same as for panels A-F but for a combination of all three *Gm*CesA isoforms. Individual and combined activities were determined at a concentration of 6.6 μM for each *Gm*CesA isoform. DPM, disintegrations per minute. In all panels, error bars represent deviations from the means of at least three replicates.

The online version of this article includes the following source data and figure supplement(s) for figure 7:

**Figure supplement 1.** Control synergistic activity assays by titrating the same *Gm*CesA isoforms.

**Figure supplement 2.** Tetrathionate-inactivated CesAs interact with another *Gm*CesA isoforms.

**Figure supplement 2—source data 1.** Raw uncropped data of western blots and Coomassie-stained PAGE gels shown in *Figure 7—figure supplement 2A–F*.

**Figure supplement 2—source data 2.** Boxed source data of western blots and Coomassie-stained PAGE gels shown in *Figure 7—figure supplement 2A-F*.

**Figure supplement 3.** Synergistic cellulose biosynthesis with tetrathionate inactivated CesA trimers.

---

*Gm*CesA1 exhibits higher in vitro catalytic activity compared to *Gm*CesA3 and *Gm*CesA6 (*Figure 2D* and *Figure 1—figure supplement 1B*). To test whether the observed synergistic effects are due to altered catalytic activity of only one isoform or both, one *Gm*CesA isoform was inactivated after purification by incubation with the oxidant sodium tetrathionate (*Tie et al., 2004*). While the inactivated CesAs exhibit activity levels comparable to EDTA-treated negative controls, they remain trimeric and interact with other isoforms as observed for the unmodified versions (*Figure 7—figure supplement 2*). Inactivation could be due to modification of a conserved cysteine residue in CesA's catalytic pocket (such as Cys558 or Cys630 in *Gm*CesA6).

Performing the above-described activity assays with pairs of inactive and active *Gm*CesA trimers demonstrates that all three isoforms exhibit increased catalytic activity when combined with an inactive trimer of a different isoform (*Figure 7—figure supplement 3*). This suggests that intertrimer interactions impact the catalytic activity of all isoforms, perhaps by altering the accessibility of the catalytic pocket (discussed below).

## Discussion

A hallmark of plant cellulose biosynthesis is the deposition of microfibrils in the cell wall (*Zhang et al., 2021b*). Prevailing models of microfibril forming CSCs postulate that they contain at least three different CesA isoforms (*Newman et al., 2013*; *Turner and Kumar, 2018*). However, experimental evidence supporting the presence of different CesA isoforms in a CSC and its repeat unit is lacking.

Despite efficient co-expression of differently tagged *Gm*CesA1, *Gm*CesA3, and *Gm*CesA6 in insect Sf9 cells, we failed to isolate hetero-oligomeric complexes suitable for structural analysis (*Figure 4—figure supplement 2*). This failure does not exclude the formation of a small fraction of hetero-oligomeric *Gm*CesA complexes; however, it indicates the preferred formation of homo-oligomeric *Gm*CesA complexes. Therefore, we characterized the *Gm*CesA isoforms individually. All three isoforms can be purified as catalytically active homotrimeric species. Trimerization is mediated by the cytosolic PCR, as observed in the secondary cell wall *Ptt*CesA8 and *Gh*CesA7 isoforms. Because the PCR is highly conserved across the three *Gm*CesA isoforms, the apparent failure (or low efficiency) of heterotrimer formation may be due to subtle differences in sequence and shape complementarity between the isoforms.

The different arrangement of TM helix 7 in the *Gm*CesAs compared to *Ptt*CesA8 and *Gh*CesA7 creates a large lateral window in the cellulose secretion channel. The window opens towards the trimer's threefold symmetry axis. While the biological function of this window or the flexibility of TM helix 7 are unclear, it could enable the lateral release of the glucan chains towards the center of the complex, thereby affecting protofibril formation (*Kurek et al., 2002*; *Purushotham et al., 2020*).

Our *Gm*CesA3 structure contrasts with the recently reported dimeric organization of a fragment of *A. thaliana's* (*At*) CesA3 catalytic domain, which self-associates via β-strand augmentation (*Qiao*

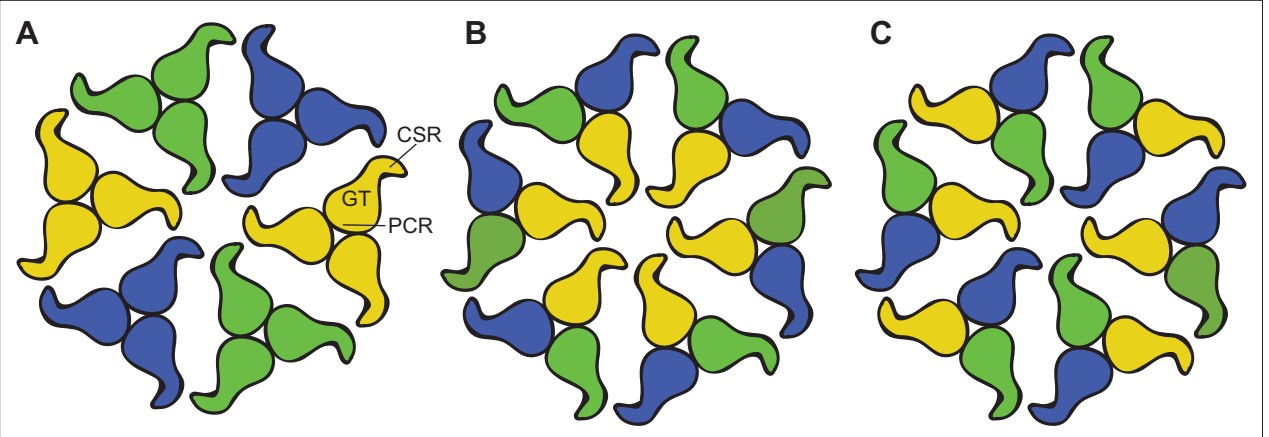

**Figure 8.** CSC models consisting of different CesA isoforms. (**A**) Association of homotrimers of three different CesA isoforms. Isoforms are indicated by different colors. The shapes represent the cytosolic CesA domains. (**B and C**) Alternative models of CSC assembly from heterotrimeric CesA complexes that have not been detected in vitro. Model (**B**) would require interactions between the same CesA isoforms at the center, also not observed in vitro. PCR: Plant conserved region, GT: Glycosyltransferase, CSR: Class specific region.

*et al., 2021*). Our cryo-EM analysis of *Gm*CesA3 and the other isoforms only revealed monomeric and trimeric states, likely because the region involved in dimerization of the *At*CesA3 fragment is inaccessible in the full-length *Gm*CesA3 protein.

Our in vitro CesA interaction studies replicate previous in vivo co-immunoprecipitation results on primary and secondary cell wall CesAs (*Gonneau et al., 2014*; *Timmers et al., 2009*). The robust interaction of trimers of different CesA isoforms supports their physiological significance. In a detergent solubilized state, the individual CesA trimers are not confined to the same plane, as is the case in a biological membrane. Thus, the in vitro observed interactions lead to clustering of the trimers, instead of their ordered close packing into symmetric particles.

Our data further demonstrate that the CSR, a region predicted to be intrinsically disordered, is the primary mediator of trimer-trimer interactions (*Figures 6 and 8*), in agreement with earlier suggestions (*Sethaphong et al., 2013*; *Singh et al., 2020*). The observation that CesA trimers can assemble in a non-physiological upside-down orientation suggests that the CSRs in a complex remain flexible, similar to previously described fuzzy complexes of intrinsically disordered proteins (*Fuxreiter, 2012*). A fuzzy CSR-CSR interface may function like a magnet to associate CesA trimers into a CSC, thereby bestowing flexibility on the CSC during microfibril formation. The nascent cellulose fiber may further stabilize the complex.

We postulate the presence of distinct interaction hotspots for two nonidentical CesA isoforms within a CSR. A dimeric complex of, for example *Gm*CesA1 and *Gm*CesA3, would then only be able to interact with *Gm*CesA6, thereby explaining the functional importance of three CesA isoforms. This model can be modified by assuming multiple binding sites for the same 'non-like' isoform or even a 'like' isoform to account for possible CSC configurations of two or one isoforms, respectively. Because trimers of the same isoform do not interact in vitro, CSC models relying on interactions between the same CesA isoforms across the repeat units are unlikely (*Figure 8*).

Due to its disordered nature (*Scavuzzo-Duggan et al., 2018*), the CSR likely occupies a large volume at the periphery of the catalytic domain. It is thus possible that the domain affects nucleotide binding to or exchange at the active site. The CSR could be repositioned upon complex formation with another isoform, which in turn could increase the catalytic activity, explaining the synergistic effects observed in vitro.

Lastly, the biological functions of the different CesA isoforms are currently unknown. Requiring different isoforms to form a functional CSC could provide regulatory control over the cellulosic material deposited in the cell wall. It is conceivable that CesA trimers function alongside fully assembled CSCs in the plasma membrane, thereby producing proto- and microfibrils (from CesA trimers and CSCs, respectively) that may interact. Accordingly, controlling the ability of the CesAs to assemble into CSCs by regulating the isoform composition in the plasma membrane may allow tailoring the fibril-to-protofibril ratio and thereby wall properties. Addressing these questions will require detailed

in vivo studies of the oligomerization and distribution of cellulose depositing CesA complexes in the plasma membrane.

## Materials and methods

### Molecular phylogeny

A pre-computed cluster of loose CesA homologues (HOM05D000074) was downloaded from Dicots PLAZA 5.0 (*Van Bel et al., 2022*), from which soy (*Glycine max*), *Arabidopsis*, cotton (*Gossypium hirsutum*), poplar (*Populus trichocarpa*), tomato (*Solanum lycopersicum*), Physcomitrium, and Amborella protein sequences were isolated. The sequence of CesA8 from *Populus tremula* x *tremuloides* (*Ptt*CesA8) was also added at this stage for later reference. A region corresponding to the Pfam 'Cellulose synthase' domain HMM profile (PF03552) was then extracted using HMMER (*Eddy, 2011*) with an *E*-value cutoff of $1 \times 10^{-80}$ and a bespoke Python script. The extracted sequences were aligned using MAFFT v7.490 (*Katoh and Standley, 2013*) before an initial near-maximum likelihood phylogeny using FastTree under default settings. This tree was used to extract *bona fide* CesA sequences from Csl sequences using Figtree. The corresponding rows of the alignment were then extracted alongside *A. thaliana* CslD5, which was used as an outgroup in the following analysis. Model selection was carried out using ProtTest version 3.4.2 (*Darriba et al., 2011*) before calculating the final maximum-likelihood phylogeny using RAxML version 8.2.12 (*Stamatakis, 2014*) under a JTT + I + Γ model with 100 rapid bootstrap pseudo-replicates. The tree was rendered in Figtree with subsequent labelling in Inkscape.

### Cloning

The primary cell wall CesA1, CesA3 and CesA6 genes from soybean (Glyma.06G069600, Glyma.12G237000, and Glyma.02G080900) were synthesized (Gene Universal) with an N-terminal 12 x His-tag coding sequence and cloned into *Not*I and *Hind*III restriction sites in the pACEBac1 vector. The N-terminally TwinStrep-tagged *Gm*CesA1 and *Gm*CesA3 constructs were generated by Quik-Change mutagenesis from the pACEBac1-12xHisCesA vectors. The N-terminally deleted *Gm*CesA constructs (ΔNCesAs) were generated by PIPE cloning from the full-length constructs, resulting in plasmids pACEBac1-ΔNCesA1R260, pACEBac1-ΔNCesA3V243, and pACEBac1-ΔNCesA6M248. The CSR regions of *Gm*CesA1 (residues 654–713) and *Gm*CesA3 (residues 648–709) were replaced with the sequence ASGAGGSEGGGSEGGTSGAT (*Baytshtok et al., 2017*) in the -ΔNCesA1R260 and -ΔNCesA3V243 backgrounds by gene synthesis.

A-multi *Gm*CesA expression cassette was generated for the co-expression of 12xHis-*Gm*CesA1, TwinStrep-*Gm*CesA3 and 3xFlag-*Gm*CesA6 using homing endonuclease/BstXI multiplication according to the protocol detailed in the MultiBac Multi-Protein Expression in Insect Cells manual (Geneva Biotech). Briefly, the TwinStrep-*Gm*CesA3-pACEBac1 vector (acceptor) was single restriction digested with BstXI. The His-*Gm*CesA1-pACEBac1vector was double restriction digested with I-CeuI and BstXI to excise the promoter and terminator containing His-*Gm*CesA1 (donor). The resulting fragment was inserted into the BstXI-digested TwinStrep-*Gm*CesA3-pACEBac1 vector to generate the TwinStrep-*Gm*CesA3_His-*Gm*CesA1-pACEBac1 construct. The above cloning procedure was repeated one more time to insert the 3xFlag-*Gm*CesA6 (donor) into the *Gm*CesA1 and *Gm*CesA3 containing vector (acceptor) to generate the multi-*Gm*CesA expression construct TwinStrep-CesA3_His-*Gm*CesA1_3xFlag-*Gm*CesA6-pACEBac1.

### Virus generation

An aliquot of 3 µL of 100 ng/µL pACEBac1-*Gm*CesA plasmid was used for transformation into 50 µL chemically competent DH10MultiBacTurbo *E. coli* cells. Bacmids were isolated from white colonies on a Bluo-Gal agar plate and transfected into *Spodoptera frugiperda* 9 (SF9) cells. P0, P1, and P2 baculovirus was generated according to the Joint Centre for Innovative Membrane Protein Technologies (JCIMPT) protocol.

### Protein expression and purification

For *Gm*CesA expression, Sf9 insect cells were infected with 15 mL P2 baculovirus per 400 mL at a density of $3 \times 10^{-6}$ cells per mL and grown at 27 °C for 48–72 hr in an orbital shaker. Cells were then

harvested by centrifugation at 5,000×*g* for 10 min at 4 °C. Cell pellets were resuspended in buffer A (20 mM Tris-HCl, pH 7.5, 100 mM NaCl, 5 mM sodium phosphate, 5 mM sodium citrate, and 1 mM TCEP) supplemented with 1% lauryl maltose neopentyl glycol (LMNG, Anatrace), 0.2% cholesteryl hemisuccinate (CHS, Anatrace), and protease inhibitors PIC (0.4 mM AEBSF, 2 µM aprotinin, 30 µM pepstatin, 7.5 µM pepstatin, 40 µM bestatin, 35 µM E-64, and 4.5 mM benzamidine hydrochloride) and lysed in a glass dounce homogenizer  (~30 strokes). The lysate was solubilized at 4 °C for 1 hr on a rocker. After separation of insoluble material by centrifugation at 200,000×*g* for 45 min, 5 mL of Ni-NTA resin (HisPur Ni-NTA Resin, Thermo scientific) and 20 mM imidazole was added to the supernatant and incubated for 1 hr at 4 °C on a rocker. After batch binding, the resin was packed into a gravity flow column and then sequentially washed twice with ten column volumes each of buffer A containing 40 mM imidazole, 0.02% glyco-diosgenin (GDN, Anatrace) and PIC, followed by ten column volumes of buffer A containing 1 M NaCl and 0.02% GDN (wash 3) and PIC. The final wash step (wash 4) was with ten column volumes of buffer A containing 0.02% GDN, PIC and 60 mM imidazole. The *Gm*CesAs were eluted with six column volumes of buffer A containing 0.02% GDN, PIC and 400 mM imidazole.

The TwinStrep-tagged *Gm*CesAs were affinity purified by incubating the membrane extract with 5 mL Strep-Tactin sepharose at 4 °C for 1 hr on a rocker. After batch binding, the resin was packed into a gravity flow column and washed as described above for Ni-NTA affinity column except the buffers lacked imidazole. The TwinStrep-tagged *Gm*CesA was eluted using 6 column volumes of buffer A containing 0.02% GDN and 5 mM desthiobiotin.

His- and TwinStrep-tagged *Gm*CesAs were further purified by size-exclusion chromatography (SEC) using Superose 6 Increase 10/300 GL column (Cytiva) equilibrated in buffer A containing 0.02% GDN without any protease inhibitors. The purified *Gm*CesA trimers and monomers were immediately used for activity assays and pulldown experiments or flash-frozen in liquid nitrogen and stored at –80 °C.

## CesA-CesA pull-down assays

After SEC, either the trimeric or monomeric fractions of His-*Gm*CesA and TwinStrep-*Gm*CesA were pooled and used for tandem affinity chromatography using Ni-NTA and Strep-Tactin sepharose beads. His-CesA1 and TwinStrep-CesA3 trimers were mixed at 150 µg/mL concentration and incubated at 4 °C for 3 hr. The mixture was first loaded onto 200 µL bed volume Ni-NTA beads for 1 hr at 4 °C in the presence of 20 mM imidazole. After collecting the flowthrough, the beads were washed with 10 bed volumes of buffer A containing 0.02% GDN and 20 mM imidazole for three washes followed by elution in 5 bed volumes of buffer A containing 0.02% GDN and 400 mM imidazole.

The Ni-NTA eluted material was next bound to 200 µL Strep-Tactin sepharose beads for 1 hr at 4 °C in a rotator shaker. After batch binding, the beads were packed into a gravity flow column. The flowthrough was collected and the column was subsequently washed three times with 10 column volumes of buffer A containing 0.02% GDN. Bound proteins were eluted with buffer A containing 0.02% GDN and 5 mM desthiobiotin. All fractions were analyzed by SDS-polyacrylamide gel electrophoresis (SDS-PAGE) and Western blotting using anti-His (QIAGEN, #34650) and anti-Strep (MilliporeSigma) primary antibodies and a DyLight 800-coupled anti-mouse secondary antibody (Rockland, #610-145-002) for detection.

The same protocol was used to study the interactions of *Gm*CesA1 and *Gm*CesA6 or *Gm*CesA3 and *Gm*CesA6 using Strep-tagged *Gm*CesA1 and His-tagged *Gm*CesA6, Strep-tagged *Gm*CesA3 and His-tagged *Gm*CesA6, respectively. Control binding experiments were performed by loading the His-tagged *Gm*CesA species onto Strep-Tactin sepharose beads and the Strep-tagged *Gm*CesAs onto the Ni-NTA beads. In each case, the beads were washed as described above.

## Inactivation of *Gm*CesA

*Gm*CesA was inactivated by treating *Gm*CesA trimers with 10 mM sodium tetrathionate overnight at room temperature (24 °C). The treated sample was purified over a size exclusion column to remove the excess sodium tetrathionate. Binding experiments with inactivated and untreated CesAs were performed using Ni-NTA beads as described above.

### *Gm*CesA activity assay

Freshly purified or aliquots of flash-frozen enzyme thawed on ice were used for activity assays. In general, activity assays were performed as described earlier (*Purushotham et al., 2020*). Radiometric quantification of in vitro synthesized cellulose was performed by combining 5 µM *Gm*CesA, 5 mM UDP-glucose (UDP-Glc), and 0.34 µM (12.5 mCi/L) UDP-[$^3$H]-Glc in buffer containing 20 mM Tris-HCl, pH 7.5, 100 mM NaCl, 20 mM MgCl$_2$, 5 mM sodium phosphate, 5 mM sodium citrate, and 1 mM TCEP. The reaction mixtures were incubated for 45min at 37°C. After incubation, the entire reaction mixture was spotted on Whatman Grade 3 MM chromatography paper. Free substrate was removed by descending paper chromatography in 60% ethanol. The radioactivity retained at the origin was quantified by scintillation counting.

The pH optima of *Gm*CesAs were determined by incubating the *Gm*CesAs in MMT buffer, consisting of DL-malic acid, MES and Tris base in the molar ratios 1:2:2-DL-malic acid:MES:Tris base. The desired pH was adjusted with NaOH or HCl. Activity assays were performed as mentioned above in technical triplicate from two biological replicates.

Cellulase digestions were performed by adding 5 U of endo-β–1,4-glucanase (*Trichoderma longibrachiatum*; Megazyme: E-CELTR) directly to the reaction mixture. Following the in vitro synthesis reaction, cellulase treatment was performed for 3 hr at 37 °C. Cellulose quantification by scintillation counting was performed as mentioned above to quantify the product.

Steady-state kinetic analyses were performed in triplicate using the UDP-Glo glycosyltransferase Assay kit (Promega) to monitor the released UDP according to the manufacturer's instructions. For measuring enzyme kinetics, the reaction mixtures containing 0.33 and 3.3 nM trimeric *Gm*CesA1 or *Gm*CesA3, respectively, and 0–1 mM UltraPure UDP-Glc (Promega) were added to 20 mM Tris-HCl buffer, pH 7.5, 100 mM NaCl, 5 mM sodium phosphate, 5 mM sodium citrate, 20 mM MgCl$_2$, 1 mM TCEP, and 0.02% GDN and incubated in a final volume of 25µL for 1 h at 30 °C. Afterwards, the reaction mixture was mixed with an equal amount of UDP-Glo reagent (Promega) in a 96-well Nunclon Delta-Treated flat-bottom microplate (Thermo Fisher Scientific) and incubated for 1 h at room temperature before measuring luminescence using a GloMax Explorer plate reader (Promega). A standard curve was used for quantification of the UDP produced. Kinetic values were obtained using the nonlinear regression function in GraphPad Prism.

### *Gm*CesA UDP inhibition assays

UDP inhibition was analyzed using radiometric quantification of in vitro synthesized cellulose, as described above by titrating 0.01–7.5 mM UDP in the reaction. Substrate concentrations for the individual reactions were 1.4 mM, 0.5 mM, and 2.3 mM for *Gm*CesA1, *Gm*CesA3, and *Gm*CesA6, respectively. Inhibition constants (IC$_{50}$) for each *Gm*CesA were obtained by data analysis in GraphPad Prism.

### Synergistic cellulose biosynthesis

Activity synergism between different *Gm*CesA isoform trimers were studied by mixing two *Gm*CesA isoforms, one at a constant concentration of 20 µM and one at increasing concentrations from 1 to 20 µM. The activities of the individual isoforms at the respective concentrations were also measured. In vitro synthesized cellulose was quantified by scintillation counting, as described above.

### EM grid preparation and data collection

After size exclusion chromatography, the freshly purified protein fractions were pooled. The protein quality was monitored by negative stain EM. The proteins were diluted to 0.01 mg/mL and 4 µL was applied to a glow discharged Formvar/Carbon grid (Electron Microscopy Sciences) for 30 s, followed by 2 x washes with 4 µL H$_2$O. The grid was negatively stained with 4 µL 0.75% Uranyl Formate (UF) in H$_2$O for 30 s. Excess UF was removed by blotting with filter paper and the grid was air dried. Images were taken on a Tecnai F20 at the Macromolecular Electron Microscopy Core (MEMC) facility at the University of Virginia.

For cryo grid preparation, the protein samples were concentrated until NanoDrop readings reached 3 mg/ml (1.8 mg/mL using an extinction coefficient of ~200,000/ (M cm)). 2.5 µL were applied to a C-flat 300 mesh 1.2/1.3 copper grids (Electron Microscopy Sciences), glow-discharged in the presence of amylamine at 25 mA for 45 s, and blotted with a Vitrobot Mark IV (FEI, Thermo Fisher Scientific) with force 4 for 6 s at 4 °C, 100% humidity, and frozen in liquid ethane.

Cryo-EM data were collected at the MEMC at the University of Virginia on a Titan Krios microscope operated at 300 keV and equipped with a Gatan K3 direct electron detector positioned post a Gatan Quantum energy filter. On average, a total of ~6000 movies were collected from one or two grids in counting mode at a magnification of 81 K, pixel size of 1.08 Å, and defocus range from –2.2 to –1.2 μm with step size of 0.2 μm. The total dose was 50 e⁻/Å². Movies with 40 frames were collected at 5.17 s/movie rate.

### Cryo-EM data processing

Cryo-EM data processing was done in cryoSPARC v4 (*Punjani et al., 2017*). The general workflow is described as supplemental information, *Figure 3—figure supplement 1*. Based on improved map qualities, all volumes were generated imposing threefold symmetry (C3). Maps generated without symmetry assignment (C1) were used to visualize the unidentified ligand coordinated by the PCR domains at the particle's symmetry axis. Initial protein models were generated in AlphaFold and manually adjusted in Chimera and Coot (*Emsley and Cowtan, 2004*; *Pettersen et al., 2004*). Models were refined in Phenix.refine (*Afonine et al., 2012*). The following regions were omitted from the constructs due to weak or missing map density: *Gm*CesA1:1–260 (NTD), 654–717 (CSR), 954–978 (gating loop), 1064–1078 (C term); *Gm*CesA3: 1–251 (NTD), 648–712 (CSR), 948–974 (gating loop), 1029–1079 (C term and TM 7); and *Gm*CesA6: 1–248 (NTD), 646–712 (CSR), 952–971 (gating loop), 1060–1078 (C term). Cellobiose was modeled at the acceptor positions of *Gm*CesA3 and *Gm*CesA6. The corresponding density in the *Gm*CesA1 map was too weak for interpretation. Structural representations were generated in ChimeraX or Pymol (*Pettersen et al., 2021*; *PYMOL, 2025*).

### Materials availability

CesA containing expression constructs are available upon request.

## Acknowledgements

We are grateful to Kelly Dryden and Michael Purdy of UVA's Macromolecular Electron Microscopy Core (MEMC) facility for support during EM data collection. This work was supported in part by BASF (CesA cloning, expression, and structure determination) and the Center for LignoCellulose Structure and Formation (CLSF, biochemical and interaction studies). CLSF is an Energy Frontier Research Center funded by the U.S. Department of Energy, Office of Science, Basic Energy Sciences (award DESC0001090). R H and J Z are supported by NIH grant R35GM144130 awarded to J Z. L W is supported by the Howard Hughes Medical Institute of which J Z is an investigator.

## Additional information

### Funding

| Funder | Grant reference number | Author |
| --- | --- | --- |
| National Institutes of Health | R35GM144130 | Ruoya Ho Jochen Zimmer |
| Howard Hughes Medical Institute | | Jochen Zimmer Louis FL Wilson |

The funders had no role in study design, data collection and interpretation, or the decision to submit the work for publication.

### Author contributions

Ruoya Ho, Pallinti Purushotham, Data curation, Investigation, Writing – review and editing; Louis FL Wilson, Validation, Investigation, Writing – review and editing; Yueping Wan, Investigation; Jochen Zimmer, Conceptualization, Data curation, Formal analysis, Writing – original draft, Project administration

### Author ORCIDs

Ruoya Ho ⓘ http://orcid.org/0000-0002-2369-8443

Pallinti Purushotham [ID] https://orcid.org/0000-0002-5565-1762
Louis FL Wilson [ID] https://orcid.org/0000-0001-6438-3328
Jochen Zimmer [ID] https://orcid.org/0000-0002-8423-2882

Reviewer #1 (Public review): https://doi.org/10.7554/eLife.96704.3.sa1
Reviewer #3 (Public review): https://doi.org/10.7554/eLife.96704.3.sa2
Author response https://doi.org/10.7554/eLife.96704.3.sa3

## Additional files

### Supplementary files
Supplementary file 1. Cryo-EM data collection, refinement, and validation statistic.
MDAR checklist

### Data availability
Protein coordinates and cryo EM maps have been deposited in PDB under the accession codes 8VHZ, 8VHT, and 8VI0.

The following datasets were generated:

| Author(s) | Year | Dataset title | Dataset URL | Database and Identifier |
|---|---|---|---|---|
| Ho R, Palliniti P, Zimmer J | 2025 | Cryo EM structure of a soybean CesA1 homotrimer | https://www.rcsb.org/structure/8VHZ | RCSB Protein Data Bank, 8VHZ |
| Ho R, Palliniti P, Zimmer J | 2025 | Cryo EM structure of a soybean CesA3 homotrimer | https://www.rcsb.org/structure/8VHT | RCSB Protein Data Bank, 8VHT |
| Ho R, Palliniti P, Zimmer J | 2024 | Cryo EM structure of a soybean CesA6 homotrimer | https://www.rcsb.org/structure/8VI0 | RCSB Protein Data Bank, 8VI0 |

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
