## [Editor Report · eLife Assessment]

It is well established that cellulose synthesis in higher plants requires three different but related cellulose synthase (CESA) isoforms. Here the authors provide **convincing** biochemical and cryo electron microscopy structural information on the interactions within soybean primary cell wall CESA homotrimers. They present an **important** model in which multi-subunit cellulose synthase complexes are made of homotrimers of different CESA isoforms.

---

## [Referee Report · Reviewer #1 (Public review)]

Cellulose is the major component of the plant cell wall and as such is a major component of all plant biomass on the planet. It is made at the cell surface by a large membrane-bound complex known as the cellular synthase complex. It is the structure of the cellulose synthase complex that determines the structure of the cellulose microfibril, the unit of cellulose found in nature. Consequently, while understanding the molecular structure of individual catalytic subunits that synthesise individual beta 1-4 glucose chains is important, to really understand cellulose synthesis it is necessary to understand the structure of the entire complex.

In higher plants cellulose is synthesised by a large membrane-bound complex composed of three different CESA proteins. During cellulose synthesis in the primary cell wall this is composed of members of groups CESA1, CESA3 and CESA6. While the authors have previously presented structural data on CESA8, required for cellulose synthesis in the secondary cell wall, here they provide structural and enzymatic analysis of CESA1, CESA3 and CESA6 from soybean.

The authors have utilised their established protocol to purify trimers for all three classes of CESA proteins and obtain structural information using electron microscopy. The structures reveal some subtle, but interesting differences between the structures obtained in this study and that previously obtained for CESA8. In particular, they identify a change in the position of transmembrane helices 7 that in previous structures formed part of the transmembrane channel. In the structure of CESA1 TM7 is shifted laterally to a position more towards the periphery of the protomer where is stabilised by inter protomer interactions. This creates a large lipid exposed channel opening that is likely encountered by the growing cellulose chain. In the discussion the authors speculate this channel might facilitate lateral movement of cellulose chains in the membrane what would allow them to associate to form the microfibril. There is, however, no explanation for why this might be different for CESA proteins involved in primary and secondary cell wall CESA proteins.

Interactions within the trimer as stabilised by the plant conserved regions (PCR), while in common with previous studies that class-specific regions (CSR) is not resolved, likely of it being highly disordered as has been suggested in previous studies. As the name suggests these regions are likely to be important for determining how different CESA proteins interact, but it remains to be seen how they achieve this. Similarly, the N-terminal domain (NTD) remains rather intriguing. In the CESA3 structure, the NTD forms a stalk that protrudes into the cytoplasm that was previously observed for CESA8, while it remains unresolved in CESA1 and CESA6. The authors suggest the inability to resolve this region is likely the result of the NTD being able to form multiple conformations. Loss of the NTD does not prevent the formation of trimers and CESA1 and CESA3 are still able to interact. Previous bioinformatic studies suggest that the CSR part of the NTD is also highly class-specific (Carrol et al. 2011 Frontiers in Plant Science 2, 5-5) suggesting it is also likely to participate in interactions between different CESA proteins. This analysis provides little new information on the structure of the NTD or how it functions as part of the cellulose synthase complex.

The other important point regarding cellulose synthesis is how the different CESA trimers function during cellulose synthesis and complex assembly. The authors provide biochemical evidence that mixed complexes of two different CESA proteins are able to synergistically increase the rate of cellulose synthesis. This increase is not dramatic, around 2-fold as it is unclear what brings about this increase and whether it results from the ability to form larger complexes favouring greater rates of cellulose synthesis.

It is clear however from electron microscopy that mixing of CESA proteins can lead to the formation of large aggregates not seen with single CESA proteins. The aggregates observed do not form rosette type shapes but appear to be much more random aggregates of different CESA trimers. The authors suggest that this is likely a result of the fact that the complexes are not constrained in two dimensions by the membrane, however if these are biologically relevant interactions that form aggregates is somewhat surprising that they do not form hexameric structures, particularly since that are essentially forming as a single layer.

Overall the study provides some important data and raises a number of important questions.

---

## [Referee Report · Reviewer #3 (Public review)]

Cellulose is a major component of the primary cell wall of growing cells and it is made by cellulose synthases (CESAs) organized into multi-subunit complexes in the plasma membrane. Previous results have resolved the structure of secondary cell wall CESAs, which are only active in a subset of cells. Here, the authors evaluate the structure of CESAs from soybean (Glycine max, Gm) via cryo-EM and compare these structures to secondary cell wall CESAs. First, they express a select member of the GmCESA1, GmCESA3, or GmCESA6 families in insect cells, purified these proteins as both monomers and homotrimers, and demonstrated their capacity to incorporate 3H-labelled glucose into cellulase-sensitive product in a pH and divalent cation (e.g., Mg2+) -dependant fashion (Figure 2). Although CESA1, CESA3, and a CESA6-like isoforms are essential for cellulose synthesis in Arabidopsis, in this study, monomers and homotrimers both showed catalytic activity, and there was more variation between individual isoforms than between their oligomerization states (i.e., CESA3 monomers and trimers showed similar activities, which were substantially different from CESA1 monomers or trimers).

They next use cryo-EM to solve the structure of each homotrimer to ~3.0 to 3.3 A (Figure 3). They compare this with PttCESA8 and find important similarities, such as the unidentified density at a positively-charged region near Arg449, Lys452, and Arg453; and differences, such as the position and relatively low resolution (suggesting higher flexibility) of TM7, which presumably creates a large lateral lipid-exposed channel opening, rather than the transmembrane pore in PttCESA8. Like PttCESA8, an oligosaccharide in the translocation channel was co-resolved with the protein structure. Neither the N-terminal domains nor the CSRs (a plant-specific insert into the cytosolic loop between TM2 and TM3) are resolved well.

Several previous models have proposed that the cellulose synthase complexes may be composed of multiple heterotrimers, but since the authors were able to isolate beta-glucan-synthesizing homotrimers, their results challenge this model. Using the purified trimers, the authors investigated how the CESA homotrimers might assemble into higher order complexes. They detected interactions between each pair of CESA homotrimers via pull down assays (Figure 4), although these same interactions were also detected among monomers (Supplemental Figure 4). Neither catalytic activity nor these inter-homotrimer interactions required the N-terminal domain (Figure 5). When populations of homotrimers were mixed, they formed larger aggregations in vitro (Figure 6) and displayed increased activity, compared to the predicted additive activity of each enzyme alone (Figure 7). Intriguingly, this synergistic behavior is observed even when one trimer is chemically inactivated before mixing (supplemental figure 7), suggesting that the synergistic effects are due to structural interactions.

The main strength of this manuscript is its detailed characterization of the structure of multiple CESAs implicated in primary cell wall synthesis, which complements previous studies of secondary cell wall CESAs. They provide a comprehensive comparison of these new structures with previously resolved CESA structures and discuss several intriguing similarities and differences. The synergistic activity observed when different homotrimers are mixed is a particularly interesting result. These results provide fundamental in vitro support for a cellulose synthase complex comprised of a hexamer of CESA homotrimers.

The main weakness of the manuscript is that the authors' evidence that these proteins make cellulose in vitro is limited to beta-glucanase-sensitive digestion of the product. Previous reports characterizing CESA structures have used multiple independent methods: sensitivity and resistance of the product to various enzymes, linkage analysis, and importantly, TEM of the product to ensure that it makes genuine cellulose microfibrils, rather than amorphous beta-glucan.

---

## [Author Response]

The following is the authors’ response to the original reviews.

**Reviewer #1 (Recommendations For The Authors):**
I can find no problems with the experiments performed in this study, but there are several results that are not easily explained. I would like to see more consideration of possible explanations. For example, one of the major differences between the the CESA structure from primary and secondary cell walls is the displacement of TM7 in the primary cell wall CESAs that leads to the formation of lipid exposed channel. Why does this vary between primary and secondary cell wall CESA proteins? Could it explain differences in the properties, such as crystallinity between primary and secondary cell wall cellulose?

At this time, the different position of TM helix 7 observed in our GmCesA structures is just an observation. We have some emerging evidence that this helix is also flexible in POCesA8 under certain conditions; however, we do not know whether this affects catalytic activity or cellulose coalescence. We have revised the text to avoid the interpretation that TM 7 repositioning is a characteristic feature of primary cell wall CesAs only.

Similarly, regarding the formation of the larger structures from mixtures of different CESA trimers. Why do they not form roseOes? Par;cularly as these appear to be forming 2-dimensional structures.

We have included additional data on the interaction between different CesA isoform trimers (Figure 6). To answer the reviewer’s ques;on, the most likely reasons for not observing closely packed roseOe-like structures are (a) steric interferences between the micelles harboring the individual CesA trimers, and (b) the lack of a stabilizing cellulose fiber. This interpretation is supported by 2D class averages of dimers of CesA1 and CesA3 trimers (now shown in Fig. 6). The class averages show an ‘upside-down and side-by-side’ orientation of the two trimers, consistent with interferences between the solubilizing detergent micelles. The implica;ons of this non-physiological arrangement are discussed in the revised manuscript. In a biological membrane, the CesA trimers are confined to the same plane in the same orientation, which is likely necessary to form ordered arrangements.

What role does the NTD play in trimer formation given its apparent very high class specificity?

We have no data suggesting any contribution of the NTD to trimer formation. Recent work on moss CesA5 and similar AlphaFold predic;ons suggest that, for some CesAs, an extreme Nterminal region can interact with the beta sheet of the catalytic domain via beta-strand augmentation. Whether this interaction can contribute to CesA-CesA interactions remains unknown.

**Reviewer #2 (Recommendations For The Authors):**
The authors provide PDB codes but not EMDB codes for the EM maps, also I would encourage the authors to upload the raw micrographs to the EMPIAR database.

The EMDB codes are shown in Table 1 and data transfer to EMPIAR is ongoing.

Page 6 line 144, the statement "All CesA isoforms show greatest catalytic activity at neutral pH" seems to contradict the data in Figure 1e and the subsequent statements. This sentence should be removed.

The text has been revised to indicate that CesA1 and CesA6 show highest activity under mild alkaline conditions.

Page 6, line 150, the authors state "The affinities for substrate binding range from 1.4 mM for CesA1 to 0.6 and 2.4 mM for CesA3 and CesA6, respectively." How were the affinities determined? Is this the affinities or the Michaelis constants? Is it known whether CesAs are rapid equilibrium enzymes? This should be clarified.

The text now states that we performed Michaelis Menten kine;cs using the ‘UDP-Glo’ glycosyltransferase assay kit. We are uncertain about whether CesAs can be classified as rapid equilibrium enzymes. The rate-limiting step of cellulose biosynthesis has been proposed to be glycosyl transfer, rather than cellulose transloca;on. To avoid any confusion, we changed the text from '…reveals Michaelis Menten constants for substrate binding of CesA1 and CesA3' to '…reveals Michaelis Menten constants for CesA1 and CesA3 with respect to UDP-Glc'.

Page 6, line 153, the authors state "CesA1's apparent Ki for UDP is roughly 0.8 mM, whereas this concentration is increased to about 1.2 to 1.5 mM for CesA6 and CesA3, respectively." From the Figure 1g legend, it appears that the authors performed additional experiments at different UDP-Glc concentrations in order to determine Ki that are not shown. This data should be included as a figure supplement as the data presented are insufficient to determine Ki (only IC50).

The UDP inhibition data show apparent IC50 values, and this has been corrected in the text. For each CesA isoform, the titration was done at one UDP-Glc concentration only.

Page 8, line 202, the authors state that TM helix 7 of the primary cell wall CesAs is more flexible "as evidenced by weaker density." The density for the TM helix 7 should be shown. If the density shown in Supplementary Figure 3 corresponds to TM helices the number of the helices should be indicated as it is not immediately obvious from the amino acid residue numbers.

The densities for TM helix 7 of all CesA isoforms are shown in Supplemental Figure 3. The helices are now labeled to orient the reader.

**Reviewer #2 (Public Review)**
The authors demonstrate via truncation that the N-terminus of the CesA is not involved in the interactions between the isoforms and propose that the CSR hook-like extensions are the primary mediator of trimer-trimer interactions. This argument would be strengthened by equivalent truncation experiments in which the CSR region is removed.

We performed the suggested experiment. We replaced the CSR in N-terminally truncated GmCesA1 and GmCesA3 with a 20-residue long linker. The resulting constructs assemble into homotrimeric complexes as observed for the wild type and only N-terminally truncated versions. However, the CSR-truncated constructs of the different isoforms do not interact with each other in vitro. Further, CSR-deleted GmCesA3 also does not interact with full-length CesA1, suggesting that two CSR domains of different isoforms are necessary for homotrimer interaction. This data is now shown as Fig. 5.

**Reviewer #3 (Recommendations For The Authors):**
Major Points(1) The authors state on Line 354 that they were unable to isolate heterotrimers, but they need to provide the data to support this claim; for example, it is important for readers to understand whether co-expression of all three CESAs leads to only homotrimers or only monomers. This information is essential to exclude model C in Figure 6.

We have revised the corresponding discussion and toned down the statement that heterotrimeric complexes did not form in our recombinant expression system. Co-expression of differently tagged secondary or primary cell wall CesAs in Sf9 cells has consistently resulted in negligible amounts of material that can be purified sequentially over different affinity matrices (corresponding to the tags on the recombinantly expressed CesAs – His, Strep, Flag). While this does not exclude the formation of a small fraction of hetero-oligomeric complexes (which could be trimers as observed in the structures or monomers interacting via their CSR regions), it demonstrates that CesAs favor the same isoform for trimer formation, rather than partnering with other isoforms. An example of such a purification is now shown as Supplemental Figure 8.

Determining whether heterotrimers are formed upon co-expression of different CesA isoforms requires high resolution structural analysis because co-purification of different isoforms can also be due to interactions between different homo-trimeric complexes, as demonstrated in this study.

While we cannot exclude that factors exist in planta that may prevent the formation of homotrimers and favor the formation of hetero-trimers, it is important to keep in mind that currently no experimental data supports the formation of hetero-trimeric complexes. Instead, our work demonstrates that existing data on CesA isoform interactions can be explained by the interaction of homotrimers of different isoforms.

(2) The evidence that the products of GmCEA1, GmCESA3, and GmCESA6 homotrimers are cellulose is that they consume UDP-glucose and produce a beta-glucanase-sensitive product. Other beta-glucans synthesized by similar GT2 family proteins (e.g. CSLDs, Yang et al., 2020 Plant Cell or CSLCs, Kim et al., 2020 PNAS) would be sensitive to this enzyme, and the product cannot truly be called cellulose unless it forms microfibrils. Previous reports of CESA activity in vitro have demonstrated that the products form genuine cellulose microfibrils rather than amorphous beta-glucan (via electron microscopy); extensively documented that the product is sensitive to beta-glucanase, but not other enzymes (e.g., callose or MLG degrading enzymes); provided linkage analysis of the product to conclusively demonstrate that it is a beta1,4-linked glucan; and documented a loss of activity when key catalytic residues were mutated (Purushotham et al., 2016 PNAS; Cho et al., 2017 Plant Phys; Purushotham et al., 2020 Science).Other GT2 characterization efforts have documented activity to similar standards (e.g. CSLDs, Yang et al., 2020 Plant Cell or CSLFs, Purushotham et al., 2022 Science Advances). At least one independent method should be provided, and the TEM of the product is necessary for readers to appreciate whether the product forms true cellulose microfibrils.

There may be some confusion regarding the nomenclature. Therefore, we revised the second sentence of the Introduction to define ‘cellulose’ as a beta-1,4 linked glucose polymer, in accordance with the ‘Essentials of Glycobiology’. This is also consistent with enzyme nomenclature as the primary product of cellulose synthase is a single glucose polymer, and not a fibril. For example, most bacterial cellulose synthases only produce amorphous (single chain) cellulose.

We show that the GmCesA products can be degraded with a beta-1,4 specific glucanase (cellulase), which demonstrates the formation of authentic cellulose. This study does not focus on the formation of fibrillar cellulose apart from suggesting a revised model for a microfibrilforming CSC.

(3) The position of isoxaben-resistant mutations implies that primary cell wall CESAs form heterotrimers (Shim et al., 2018 Frontiers in Plant Biology). Indeed, in their previous description of the POCESA8 structure (Purushotham et al., 2020 Science), the authors discussed the position of isoxaben-resistant mutations as a way to justify the way that TM7 of one CESA can contribute to forming the cellulose translocation pore in the neighbouring CESA within a heterotrimer. However, in this manuscript, the authors document a different location for TM7 in the GmCEA1, GmCESA3, and GmCESA6 homotrimers, which would change the position of these resistance mutations. Please discuss.

As stated in the manuscript, we do not know what the functional implication of the TM7 flexibility may be, but we speculate that it could affect the alignment of the synthesized cellulose polymers. Regarding the previously reported POCesA8 structure, the mapping of one of the reported isoxaben resistance mutants to the C-terminus of TM7 was not used to justify the structure; the structure with its position of TM7 stands on its own. Considering recent observations suggesting that isoxaben may affect cellulose biosynthesis via secondary effects, we prefer not to speculate on the mechanism by which these mutations cause the apparent resistance to isoxaben (PMID: 37823413).

(4) The authors present no evidence that GmCESA1/3/6 are involved in primary cell wall synthesis. Please include gene expression information (documenting widespread expression consistent with primary CESAs) and rigorous molecular phylogenetic analysis (or references to these published data) to clarify that these are indeed primary cell wall CESAs.

This has been addressed. We have included additional figures (Fig. 1 and S1B) that show the strong and wide distribution of the selected CesAs in soybean leaves, their co-expression with primary cell wall markers, and their phylogenetic clustering with Arabidopsis primary cell wall CesAs.

(5) Several small changes need to be made to the abstract to ensure that it aligns with the data: Line 28: add "in vitro" arer "their assembly into homotrimeric complexes" Line 28: change "stabilized by the PCR" to "presumably stabilized by the PCR".

We inserted ‘in vitro’ as requested. We did not insert the second modification as requested since CesA trimers are stabilized by the PCR. This is a fact arising from several experimentally determined CesA trimer structures.

(6) In all graphs in all figures it is unclear what the sample size is and what the bars represent. These must be stated in the figure legends. It is best practice to plot individual data points so that readers can easily interpret both the sample size and the variation.

The sample sizes and error bars are now defined in the relevant figure legends.

(7) The methods need to unambiguously define GmCESA1, GmCESA3, GmCESA6 protein identities using appropriate accession numbers.

The accession codes are now provided in the Methods.

Minor Points(1) Does CESA1 have higher activity in Figure 1D because of the pH at which the assay was conducted (see Figure 1E)? Could this difference in activity or pH preference have also affected their capacity to resolve TM7 of CESA1?

We consistently observe higher in vitro catalytic activity of CesA1, compared to CesA3 and CesA6. Activity assays are performed at a pH of 7.5, roughly halfway between the activity maxima of CesA3 and CesA1/6. At this pH, we expect activity differences to arise from factors other than the buffer pH. As detailed above, we do not know whether the conformational flexibility of TM helix 7 affects catalytic activity.

(2) Line 55: The authors should cite additional papers that also provide insight into CESA structure (e.g. Qiao et al 2021 PNAS).

A recent publication on moss CesA5 has been included. Qiao et al unfortunately report on a dimeric assembly of a fragment of *Arabidopsis thaliana’s* CesA3 catalytic domain, which we consider non-physiological. We added a brief statement in the Discussion explaining that our GmCesA3 structure is inconsistent with the dimeric arrangement reported by Qiao et al.

(3) Line 95: these references are about secondary cell wall CESA isoforms, but there are more appropriate references for the primary CESAs that should be included in place of these papers.

Fagard et al report on growth defects in roots and dark-grown hypocotyls linked to Arabidopsis CesA 1 and CesA6, which are primary cell wall CesAs. Nevertheless, we have included two additional recent publications from the Meyerowitz and Persson labs.

(4) Line 121-122: Please cite a specific figure that supports this claim, since the (Purushotham et al., 2020) reference refers to POCESA8 enrichment results, but the claims are about the GmCESA1/3/6 enrichment.

The POCesA8 reference has been removed. The classification into monomers and trimers arises from the data processing described in this manuscript and is consistent with similar results obtained for POCesA8.

(5) Line 314: It is more appropriate to use "enzyme activity" rather than "cellulose synthesis".

We prefer to use cellulose biosynthesis since the enzyme produces cellulose.

(6) Figure 1: please add colour to the graphs to clarify which trend lines belong to which data series (especially Figure 1G).

The figure (now Fig. 2) has been revised as suggested.

(7) Figure 2D: It's not clear which parts are GmCESA and which are POCESA8; please clarify the figure legend.

Thank you, the legend has been revised accordingly (now Fig. 3).

(8) In Figure 5, It's not clear that the one CESA is maintained at a steady concentration throughout the assay since there is only a bar for that CESA at the highest concentration (e.g. in Figure 5A, the blue bar for CESA1 only appears on the right-most assay, but there was CESA1 in all assays, so this should be indicated).

In the panel the reviewer is referring to, the blue bar corresponds to the activity measured for only CesA1 at a concentration of 20 µM. The red columns (indicated as ‘Mix’) represent the activities measured in the presence of 20 µM of CesA1 plus increasing concentrations of CesA3. The purple columns represent activities obtained for only CesA3 at the indicated concentrations. Numerical addition of the activities of CesA1 alone at 20 µM (blue column) and CesA 3 alone (purple columns) gives rise to the gray columns, now indicated by a capital ‘sigma’ sign. We are unclear on how the figure could be improved, but we have revised the legend to avoid confusion.

(9) Figure 5 legend needs to be clarified to indicate whether monomers or homotrimers were used in the assays.

This is now shown as Fig. 7 and the legend has been revised as requested. The experiments were performed with the trimeric CesA fractions.

(10) There seem to be some random dots near the top of Figures 6B & 6C

Removed. Thank you.